# Low N$_2$O and variable CH$_4$ fluxes from tropical forest soils of the Congo Basin

Matti Barthel [1✉], Marijn Bauters [2], Simon Baumgartner [1,3], Travis W. Drake [1], Nivens Mokwele Bey[4], Glenn Bush[5], Pascal Boeckx [2], Clement Ikene Botefa[4], Nathanaël Dériaz[1], Gode Lompoko Ekamba[4], Nora Gallarotti [1], Faustin M. Mbayu[6], John Kalume Mugula[7], Isaac Ahanamungu Makelele[2], Christian Ekamba Mbongo[8], Joachim Mohn [9], Joseph Zambo Mandea[5], Davin Mata Mpambi[4], Landry Cizungu Ntaboba[10], Montfort Bagalwa Rukeza[11], Robert G. M. Spencer[12], Laura Summerauer [1], Bernard Vanlauwe[13], Kristof Van Oost[3], Benjamin Wolf[14] & Johan Six[1]

Globally, tropical forests are assumed to be an important source of atmospheric nitrous oxide (N$_2$O) and sink for methane (CH$_4$). Yet, although the Congo Basin comprises the second largest tropical forest and is considered the most pristine large basin left on Earth, in situ N$_2$O and CH$_4$ flux measurements are scarce. Here, we provide multi-year data derived from on-ground soil flux ($n = 1558$) and riverine dissolved gas concentration ($n = 332$) measurements spanning montane, swamp, and lowland forests. Each forest type core monitoring site was sampled at least for one hydrological year between 2016 - 2020 at a frequency of 7-14 days. We estimate a terrestrial CH$_4$ uptake (in kg CH$_4$-C ha$^{-1}$ yr$^{-1}$) for montane (−4.28) and lowland forests (−3.52) and a massive CH$_4$ release from swamp forests (non-inundated 2.68; inundated 341). All investigated forest types were a N$_2$O source (except for inundated swamp forest) with 0.93, 1.56, 3.5, and −0.19 kg N$_2$O-N ha$^{-1}$ yr$^{-1}$ for montane, lowland, non-inundated swamp, and inundated swamp forests, respectively.

[1] Department of Environmental Systems Science, ETH Zurich, Zurich, Switzerland. [2] Isotope Bioscience Laboratory, Department of Green Chemistry and Technology, Ghent University, Ghent, Belgium. [3] Earth and Life Institute, Université Catholique de Louvain, Louvain-la-Neuve, Belgium. [4] Institute Congolais pour la Conservation de la Nature, Mbandaka, Democratic Republic of Congo. [5] Woodwell Climate Research Center, Falmouth, MA, USA. [6] Faculté de Gestion de Ressources, Naturelles Renouvelables, Université de Kisangani, Kisangani, Democratic Republic of Congo. [7] Département de Biologie, Université Officielle de Bukavu, Bukavu, Democratic Republic of Congo. [8] Coordination Provinciale de l'environnement, Mbandaka, Democratic Republic of Congo. [9] Laboratory for Air Pollution/Environmental Technology, Empa, Switzerland. [10] Université Catholique de Bukavu, Bukavu, Democratic Republic of Congo. [11] Département de Géodésie et Télédétection des Risques Naturels, Observatoire Volcanologique de Goma, Goma, Democratic Republic of Congo. [12] Florida State Universtity, Tallahassee, FL, USA. [13] International Institute of Tropical Agriculture, Nairobi, Kenya. [14] Division of Atmospheric Environmental Research (IFU), Karlsruhe Institute of Technology (KIT), Institute of Meteorology and Climate Research (IMK), Garmisch-Partenkirchen, Germany. ✉email: mbarthel@ethz.ch

Forest soils play a major role in the biosphere–atmosphere exchange of methane ($CH_4$) and nitrous oxide ($N_2O$), the second and third most important greenhouse gases, respectively, after carbon dioxide ($CO_2$). Tropical forests in particular are considered to be one of the main natural terrestrial sources of $N_2O$[1,2] and major sinks for atmospheric $CH_4$[3]. While strong uptake rates for $CH_4$ can mostly be attributed to diffusion, facilitated by the coarse texture of highly weathered tropical lowland forest soils[3], high $N_2O$ emissions are mostly attributed to excess nitrogen relative to phosphorus in these soils[4]. Within the tropics, Congo Basin forests have been postulated, through process-based modeling, as a major hotspot for $N_2O$ emissions[5]. At this scale, such transitive emission modeling is highly consequential, since the Congo Basin comprises the second largest contiguous tropical forest on Earth which blankets approximately half of the 3.6 million $km^2$ total basin area. However, despite its likely importance for the global $CH_4$ and $N_2O$ budget, in situ $N_2O$ and $CH_4$ flux measurements from the Congo Basin are limited. Since central African forests have been shown to differ fundamentally from other tropical biomes in terms of species richness[6,7], structure, biomass[8], and carbon uptake[9], ecological and biogeochemical conclusions cannot simply be inferred from elsewhere. In addition, the Congo Basin is subjected to a uniquely high fire-derived N deposition load[10], which might impose large ramifications on soil-atmosphere $N_2O$ and $CH_4$ exchange by potentially stimulating $N_2O$ production while increasing $CH_4$ uptake[11]. Consequently, constraining soil-atmosphere $N_2O$ and $CH_4$ fluxes through on-ground observations is paramount to adequately partition sources of global atmospheric $N_2O$ and $CH_4$, especially in data-poor but supposedly hotspot-regions like the Congo Basin.

To the best of our knowledge, only one study has reported terrestrial in situ $CH_4$ data from the Congo Basin[12] and there are no data for $N_2O$. This single study is complemented by only a few $N_2O$ and $CH_4$ measurements from African tropical forest soils outside the Congo Basin (Republic of Congo[13]; Cameroon[14–16]; Kenya[17–20]; Tanzania[21]; Ghana[22]; Uganda[23]), further highlighting central Africa as a blind spot for empirical flux measurements (Fig. 1). Consequently, this study aims to address the high uncertainty surrounding the Congo Basin's $N_2O$ and $CH_4$ budget. To achieve this, we quantified seasonal soil–atmosphere

fluxes of $N_2O$ and $CH_4$ from the three major tropical forest types (montane, lowland, swamp) within the Congo Basin at a weekly to fortnightly resolution using the static chamber method[24]. Initial short-term campaigns with daily measurements were conducted in 2016 and 2017 at three lowland locations (Maringa-Lopori-Wamba Landscape, Yangambi Biosphere Reserve, Yoko Forest Reserve) and one montane location (Kahuzi-Biéga National Park) to constrain spatiotemporal variation (Fig. S1). Expanding on these short-term expeditions, long-term observation 'core sites' were established in Yoko Forest Reserve and Kahuzi-Biéga National Park, as lowland and montane representatives, starting in 2016 and 2017, respectively. These two long-term core sites were subsequently expanded by the addition of a seasonally inundated swamp forest core site in 2019 (Jardin Botanique d'Eala). Including the initial short-term campaigns, 1558 individual soil–atmosphere flux measurements were undertaken for this study (Table S1). During the short-term campaigns, we also sampled $N_2O$ isotopes at all five sites to explore the underlying $N_2O$ source processes ($n = 63$). Lastly, as aquatic ecosystems are increasingly considered as potential emission hotspots within the terrestrial landscape, we also report dissolved $N_2O$ and $CH_4$ concentrations from headwater streams draining the same catchments in which soil flux core sites were located. Stream samples were collected for one year together with the soil flux measurements. Headwater streams and low-order tributaries in particular are believed to be important outlets for ecosystem gaseous losses[25]. Fluxes previously measured from the Congo River main stem[26] likely underestimate total aquatic losses by a significant margin, since relatively high fluxes have been observed to take place higher in the watershed[27].

## Results and discussion

**Fluxes of $N_2O$ and $CH_4$ from Congo Basin soils.** For both $CH_4$ and $N_2O$, there was considerable spatial (between chambers within sites; Table S2) and temporal (within site, across weeks; Table S2) variance, as is common for both gas species. However, even though soil moisture and temperature are known drivers of $N_2O$ and $CH_4$ fluxes, the combined variability explained by water-filled pore space (WFPS) and soil temperature was very low. In the lowland forest, WFPS and temperature only explained

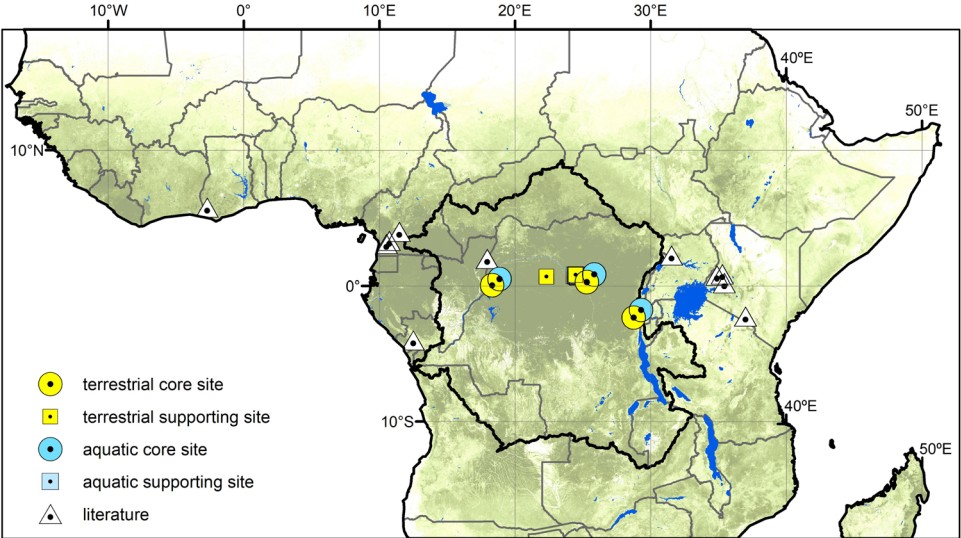

**Fig. 1 Forest cover of sub-Saharan Africa.** 2019 forest cover of sub-Saharan Africa (0% white, 100% dark green)[78] with Congo Basin boundary delineation in black showing all currently published in situ African tropical forest studies on $N_2O$ and/or $CH_4$ in white triangles[12-23] and this study in yellow (terrestrial) and blue (aquatic). Circles represent core long-term observation sites and squares represent supporting sites. Riverine dissolved $CH_4$ and $N_2O$ samples were taken from headwater streams draining the same catchments in which the core sites were located.

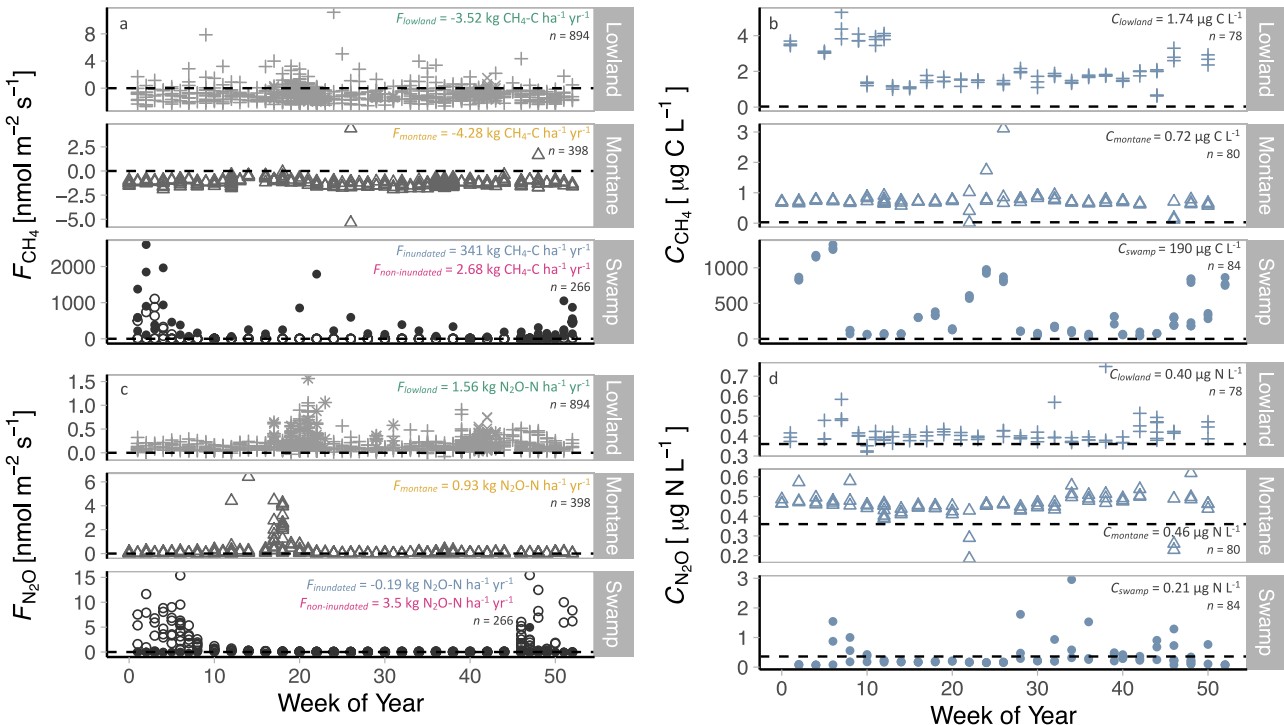

**Fig. 2 Spatiotemporal variation of terrestrial and aquatic N₂O and CH₄. a** Spatiotemporal variation of soil CH₄ fluxes measured at three different tropical forest types in the DR Congo between 2016 and 2020 and **b** corresponding dissolved CH₄ measured in headwater streams draining the soil flux core site catchment. Dashed lines indicate CH₄ at equilibrium with the atmosphere (0.03 μg C L⁻¹). **c** Spatiotemporal variation of soil N₂O fluxes measured at three different tropical forest types in DR Congo between 2016 and 2020 and **d** corresponding dissolved N₂O measured in headwater streams draining the soil flux core site catchment. Dashed lines indicate N₂O at equilibrium with the atmosphere (0.36 μg N L⁻¹). Montane forest (Kahuzi-Biéga National Park (triangles)), lowland forest (Maringa-Lopori-Wamba Landscape (star), Yangambi Biosphere Reserve (cross symbols), Yoko Forest Reserve (plus symbols)), swamp forest (Jardin Botanique d'Eala (closed circles inundated, open circles non-inundated)). Note, for visualization purposes data was pooled into weekly classes irrespective of sampling year.

5% and <1% of N₂O and CH₄ flux variability, respectively. In the swamp and montane forests, they explained slightly more but still marginal proportions of flux variability (swamp = N₂O 29%; CH₄ 26%; montane = N₂O 27%; CH₄ 33%). These results suggest that the changes of both N₂O and CH₄ fluxes within a forest type are driven by factors other than soil temperature and WFPS, which is not surprising given the low temporal variation of soil temperature and WFPS within a given location (Fig. S2).

CH₄ fluxes ranged from strong uptakes of −5.36 nmol m⁻² s⁻¹ at the montane site to a massive maximum release of 2608.12 nmol m⁻² s⁻¹ at the inundated swamp forest with a variable geometric mean [95% CIs] flux depending on forest type (Fig. 2a). While montane and lowland forests soils sequestered −4.28 [−4.70, −3.80] and −3.52 [−4.29, −2.68] kg CH₄-C ha⁻¹ yr⁻¹, respectively, the swamp forest site, under both flooded and dry conditions, was a CH₄ source, releasing 2.68 [−3.03, 12.09] from non-inundated and 341 [147, 798] kg CH₄-C ha⁻¹ yr⁻¹ from inundated areas. It is important to note, however, that the non-inundated swamp forest area was a CH₄ sink for most of the study period but the geometric mean was strongly influenced by sporadic high emissions events, as can be seen from the density distribution functions (Fig. S3). One erratic high CH₄ flux was also observed for a single chamber at the lowland site (37.6 nmol m⁻² s⁻¹, not shown) which was likely caused by transient termite activity, known to result in anomalously high CH₄ emissions[14].

Lowland forest uptake rates (−3.52 kg CH₄-C ha⁻¹ yr⁻¹) agree well with previous studies in African tropical forests (Cameroon, −3.54 kg CH₄-C ha⁻¹ yr⁻¹)[14] but are higher compared to the tropical forest world average (−2.50 kg CH₄-C ha⁻¹ yr⁻¹)[3].

Similarly, montane uptake rates (−4.28 kg CH₄-C ha⁻¹ yr⁻¹) are higher than in other Afromontane tropical forest studies with estimates of −3.2[19] and −3.12 kg CH₄-C ha⁻¹ yr⁻¹ [21]. Overall, the measured CH₄ uptake of lowland and montane study soils fell generally toward the high end of the range of other tropical montane and lowland forest sites (Veldkamp et al., 2013 and references therein). Our inundated swamp forest flux range (0−9870 kg C-CH₄ ha⁻¹ yr⁻¹) is higher than the range of CH₄ fluxes reported from an inundated forest also located in the *Cuvette Centrale* (27−1506 kg C-CH₄ ha⁻¹ yr⁻¹)[12], which is to our knowledge the only soil CH₄ study done previously within the confines of the Congo Basin. Despite the swamp forest covering only ~7% of the total humid forest area in the Congo Basin, its massive CH₄ flux renders the Congo Basin forests a source, releasing a forest type weighted average of 8.38 [0.89, 25.03] kg CH₄-C ha⁻¹ yr⁻¹ to the atmosphere if we assume our sites to be representative of their respective forest types (weighted flux based on forest coverage[28]: 90.6% lowland, 6.8% swamp, 2.6% montane). The result that Congo Basin's forests might be on average a net source of CH₄, even though 93% of its area was identified as a sink (at least when considering forest type as the sole indicator), highlights the source strength of the swamp CH₄ flux relative to the uptake measured in the other forest types. We would like to stress, however, that these extrapolation estimates should not be overinterpreted, as higher spatiotemporal data coverage is needed to draw basin-wide conclusions as to whether the Congo Basin's forests act as a net sink or source for CH₄—especially considering yet unaccounted but potentially sizeable sources such as CH₄ fluxes from swamp forest tree stems[29].

Congo Basin soil $N_2O$ fluxes ranged from $-0.22$ to $15.46$ nmol m$^{-2}$ s$^{-1}$ across all forest types and were predominantly a source to the atmosphere (Figs. 2c and S4). We determined geometric mean [95% CIs] $N_2O$ fluxes of 0.93 [0.6, 1.29] kg $N_2O$-N ha$^{-1}$ yr$^{-1}$ from montane, 3.5 [1.85, 6.34] kg $N_2O$-N ha$^{-1}$ yr$^{-1}$ from non-inundated swamp forests, $-0.19$ [$-0.75$, 0.76] kg $N_2O$-N ha$^{-1}$ yr$^{-1}$ from inundated swamp forests, and 1.56 [1.25, 1.93] kg $N_2O$-N ha$^{-1}$ yr$^{-1}$ from lowland forests. Our estimated basin-wide forest type weighted average of 1.55 [1.19, 2.02] kg $N_2O$-N ha$^{-1}$ yr$^{-1}$ was 32% higher compared to 1.17 kg $N_2O$-N ha$^{-1}$ yr$^{-1}$ reported for Cameroonian tropical forest[15] but 12–61% lower than the global averages for tropical forests based on a meta-analysis (2.8[22]; 2.0[30] kg $N_2O$-N ha$^{-1}$ yr$^{-1}$) or modeling efforts (1.76–4 kg $N_2O$-N ha$^{-1}$ yr$^{-1}$)[5].

**Isotopic evidence of $N_2O$ reduction.** To further investigate the mechanisms behind the unexpectedly low $N_2O$ emission, we analyzed the isotopic composition ($\delta^{18}O$ and $\delta^{15}N$) of emitted $N_2O$ during the intensive campaigns at all forest locations (Table S1). Isotopes of N in $N_2O$ exhibited an average site preference value of $7.22 \pm 8.5$‰, indicative of denitrification as the major driver for $N_2O$ production[31,32]. We further compared $\delta^{18}O$ and bulk $\delta^{15}N$ of $N_2O$ from this study to previous terrestrial ($n = 4$) and riverine ($n = 1$) tropical forest studies using an isotope mapping approach (Fig. 3)[33]. This mapping approach confirmed denitrification as the dominant $N_2O$ production process, with all delta values falling close to an empirical reduction vector with a ratio of 2.4 ($\delta^{18}O/\delta^{15}N$). In other words, $\delta^{18}O$ and $\delta^{15}N$ co-vary during reduction of $N_2O$ to $N_2$, which isotopically enriches the remaining soil $N_2O$[33,34]. Apart from one study in the Amazonas[35], all tropical forest sites have so far shown relatively high enrichment of both $^{18}O$ and $^{15}N$, indicating $N_2O$ reduction as a dominant process across the tropics. It should be noted that during denitrification (given limited $NO_3^-$ supply), a concurrent enrichment of $^{15}N$ and $^{18}O$ also occurs in the $NO_3^-$ substrate pool[36], resulting in a strong correlation of $^{18}O$-$NO_3^-$ and $^{15}N$-$NO_3^-$, which in turn directly translates to an increase in $\delta^{18}O$-$N_2O$ and $\delta^{15}N$-$N_2O$[37]. Consequently, high denitrification rates ultimately result in enriched $N_2O$, either through substrate enrichment and/or through reduction to $N_2$. With respect to tropical ecosystems, recent work from Costa Rica[38] suggest also weak gaseous N losses as $N_2O$ and attributed the discrepancy

between high N inputs and low outputs to $N_2$ losses. Similarly, N budgeting from tropical forest watersheds using natural abundance stable isotopes of $NO_3^-$ assigned 24–53% of the total gaseous N lost to denitrification with a high proportion of $N_2$[39]. Analogously high $N_2$ and comparatively low $N_2O$ losses during denitrification were further suggested for tropical forests using a similar approach[40] and modeled generally for humid tropical forest soils and hydrological flow paths in the range of >45% of total watershed N losses[41]. Overall, our isotopic data suggest that the proportion of $N_2O$ reduction to $N_2$ might be one driving factor behind the 10–60% lower $N_2O$ fluxes observed in the forests of the Congo Basin compared to the global tropical forest average.

**Linking terrestrial (vertical) and aquatic (lateral) $N_2O$ and $CH_4$.** Our estimates of terrestrial $N_2O$ fluxes and their isotopic signature, indicative of reduction to $N_2$, are corroborated by low dissolved $N_2O$ in headwater streams of the catchments draining our study sites in the forests (Fig. 2d). We found low riverine dissolved $N_2O$ concentrations of $0.46 \pm 0.06$ µg N L$^{-1}$ (montane), $0.40 \pm 0.06$ µg N L$^{-1}$ (lowland), and $0.21 \pm 1.16$ µg N L$^{-1}$ (swamp) with a $\delta^{15}N$ signature of $6.0 \pm 3.1$‰ (montane, lowland), indicating near equilibration with the atmosphere ($N_2O_{eq} = 0.36$ µg N L$^{-1}$ [42]; $\delta^{15}N_{eq} = 7.3$‰[43,44]). This suggests that negligible quantities of $N_2O$ are dissolved in soil pore water and laterally exported to aquatic systems and supports the assumption that exclusive measurement of vertical fluxes from soils does not underestimate total landscape $N_2O$ fluxes. At the same time, the near-atmospheric $N_2O$ concentrations and isotopic signatures found in the stream waters also rule out hydrological mineral N loss, which may be emitted further downstream as well. Our headwater stream results correspond well with earlier studies that also observed $N_2O$ in surface waters of the Congo Basin to be nearly in equilibrium with the atmosphere[26,45–47] or even undersaturated relative to the atmosphere in rivers draining swamp forests, with in-stream $N_2O$ reduction as the suggested mechanism[45,47].

In contrast to $N_2O$, headwater streams showed $CH_4$ supersaturation at all sites ($0.72 \pm 0.33$ µg C L$^{-1}$ (montane), $1.74 \pm 1.05$ µg C L$^{-1}$ (lowland), and $190 \pm 373$ µg C L$^{-1}$ (swamp)) relative to atmospheric equilibrium ($0.03$ µg C L$^{-1}$). Moreover, $CH_4$ concentrations exhibited relatively low seasonal variability

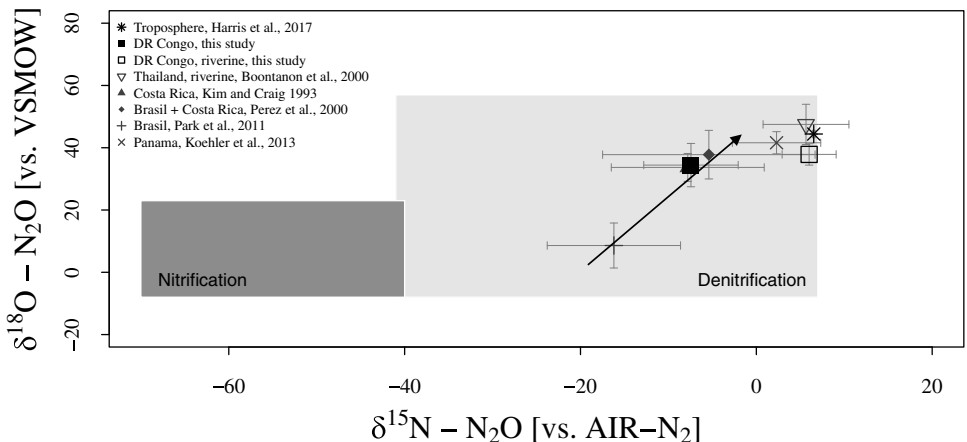

**Fig. 3 Stable isotope signatures of tropical forest soil $N_2O$ fluxes.** Isotope map of $\delta^{18}O$ vs. $\delta^{15}N$ adapted from Koba et al. (2009)[33] of tropical forest soil $N_2O$ fluxes[35,79–81] and riverine $N_2O$[82] derived from literature and this study with overall site medians given in ‰. Note, values from Koehler et al. (2012) and Pérez et al. (2000) are fully or partly from soil pore air measurements. Arrow indicates empirical reduction vector with the ratio of 2.4 ($\varepsilon^{18}O/\varepsilon^{15}N$)[33]. Rectangles depict literature ranges of $\delta^{18}O$ vs. $\delta^{15}N$ for the $N_2O$ production from nitrification and denitrification. Asterisk (*) represents isotope composition of tropospheric air as reference[44]. Error bars indicate standard deviation.

(Fig. 2b, Fig. S5a) but high spatial heterogeneity (Fig. S5a). Supersaturation of riverine $CH_4$ is commonly found in the Congo Basin[26,45,47] with subsequent outgassing partly offsetting terrestrial uptake. Our negative soil $CH_4$ fluxes at the lowland and montane site together with supersaturated dissolved $CH_4$ results suggest that aquatic $CH_4$ production is decoupled from these terrestrial ecosystems and likely occurs within benthic sediments and/or riparian zones of the streams themselves. However, high hydrological connectivity was indicated at the swamp forest site where high soil to atmosphere $CH_4$ fluxes from the inundated forest were corroborated by highly supersaturated dissolved $CH_4$ concentrations found in riverine samples ($190 \pm 373 \, \mu g \, C \, L^{-1}$). Such high fluvial-wetland connectivity driving variations of riverine greenhouse gas concentrations has been recently shown for the entire Congolese *Cuvette Centrale* based on data from 10 expeditions across the Congo River network[47].

It is important to note that in-stream gas concentrations are a result of a multitude of simultaneous processes and drivers (i.e. gas solubility, aquatic-terrestrial connectivity, inputs from surface water/groundwater, in-stream metabolism, stream chemistry, stream morphology, gas transfer velocity) which render $CH_4$ and $N_2O$ highly variable at low spatiotemporal scales. Thus, although the in-stream concentrations we observed are consistent with minimal vertical $N_2O$ losses, decoupled $CH_4$ fluxes in the lowland and montane, and strong hydrologic connectivity in the swamp forest, they do not necessarily reflect lateral soil export from the location in the catchment where soil–atmosphere fluxes were measured. We, therefore, caution that such terrestrial–aquatic linkage is likely associated with high uncertainty.

**Towards a more nuanced tropical paradigm.** Classically, tropical forests are believed to be N-rich systems in which supply exceeds biological demand because high rates of inorganic losses to the atmosphere and aquatic systems have been observed. This has led to the conclusion that tropical forests are ubiquitously strong sources of $N_2O$ (Hedin et al., 2009[48] and references therein). However, recent studies from the Congo Basin[49,50] and other data on denitrification losses from the Neotropics[39] challenge the paradigm that tropical forests are universal hotspots of $N_2O$ emissions. Instead, these studies rather indicate that a fraction of gaseous N losses occur as $N_2$ emissions. Our results align with these recent findings and suggest that the Congo Basin is not a hotspot for $N_2O$ and releases less $N_2O$ compared to other tropical forests. Contrary to reduced $N_2O$ emissions, the Congo Basin shows a higher net $CH_4$ uptake from forest soils when related to other non-inundated tropical forests. However, the sizeable $CH_4$ flux of the swamp forests together with highly concentrated riverine $CH_4$ concentrations indicate that swamp forests may offset the areally dominant $CH_4$ sink of the Congo Basin despite swamp forest covering only 7% of the humid forest and flooded areas covering only 10 % of the total basin area[51]. This is an especially important finding given that inland waters are increasingly recognized as important greenhouse gas outlets within the terrestrial landscape[26,52–54] with recent data suggesting that Congo Basin's inland waters might emit similar or even greater amounts of carbon per area than the Amazon Basin[55].

Generally, there are strong links between methanotrophic activity and the nitrogen cycle[56] with stimulatory effects on $CH_4$ uptake in response to nitrogen addition ($<100 \, kg \, N \, ha^{-1} \, yr^{-1}$) according to a meta-analysis on non-wetland soils[57]. In light of the recently discovered high fire derived N deposition on Central African forests[10] such N inputs could be one of the driving factors of the observed high biological $CH_4$ oxidation in lowland and montane forest soils. The rapidly expanding database of genome

sequences will help to unravel the biogeochemical link of $CH_4$ and $N_2O$, which has already led to interesting new discoveries such as bacteria capable of both $CH_4$ oxidation and denitrification-mediated $N_2$ production[58]. It is possible that such microbial groups might play a role in the soils of Congo's tropical forests given both the high methanotrophic activity and signatures of complete denitrification identified in this study.

Overall, our measurements add to key studies on African tropical forests[6–9,59], highlight the role of the Congo Basin as a global greenhouse gas buffering reservoir, and stress once again its value in global conservation efforts. Lastly, we show that on-ground observational efforts remain imperative for improved quantification of regional and global greenhouse gas balances. While dense greenhouse gas observational infrastructures have been established at continental scales (e.g. ICOS, NEON) to feed global databases (e.g. FLUXNET[60]), ground-based observations from the African continent are still severely underrepresented[61–63] as can be seen from poor sub-Saharan coverage of soil–atmosphere specific greenhouse gas flux databases (COSORE[64], Global $N_2O$ Database[65]). As a consequence, the contribution of African tropical forests (and African ecosystems in general) to global atmospheric exchange of $CH_4$ and $N_2O$ remains subject to high uncertainty, despite the fact that $CH_4$ and $N_2O$ emissions are thought to shift the African continent as a whole to a net source of greenhouse gases[62]. Our study is the first step towards addressing this uncertainty and highlights both the variation and nuanced nature of these key biogeochemical fluxes.

## Methods

**Sites.** All field sites are located in the Democratic Republic of Congo. Three long-term observation core sites were established, one in the montane forest of the Albertine Rift (Kahuzi-Biéga National Park, South Kivu province), one in a pristine lowland forest 30 km south of the city of Kisangani (Yoko Forest Reserve, Tshopo province), and one in a seasonally flooded swamp forest 7 km away from the town of Mbandaka (Jardin Botanique d'Eala, Équateur province). The montane forest climate is classified as Cfb-type following the Köppen-Geiger classification, with *Allophylus kivuensis*, *Maesa lanceolata*, *Lindackeria kivuensis*, *Bridelia micranta*, and *Dombea goetzenii* as the main occurring tree species at this altitude[66]. The montane forest soils are classified as Umbric Ferrasols with a sandy loam texture[67]. The lowland forest climate is classified as Af-type at the Köppen–Geiger classification with highly weathered, nutrient-poor, acidic, and loamy sand textured Xanthic Ferrasol soils[68]. The main occurring species at the core lowland forest site are *Scorodophloeus zenkeri*, *Julbernardia seretii* (De Wild.) Troupin, *Prioria oxyphylla* (Harms) Breteler, *Cynometra hankei*, *Tessmannia africana* with mono-dominant forest patches of *Gilbertiodendron dewevrei* (De Wild.) J. Léonard[69]. The seasonally inundated swamp forest core site investigated was located within a botanical garden operated by the *Institute Congolais pour la Conservation de la Nature* (ICCN). The botanical garden comprises 371 ha of land consisting of 35% dense swamp forest, 14% forest on firm ground, 32% open forest, and the remaining area consisting of secondary forest, grassland, and deforested land of which 189 ha are protected forest area. The main tree species found at the swamp forest site are *Hevea brasiliensis*, *Ouratea arnoldiana*, *Pentaclethra eetveldeana*, *Strombosia tetandra*, and *Daniella pynaertii*. The soil at the site was characterized as Eutric Gleysols with a sandy loam texture.

Before the establishment of the long-term observation core sites, two additional lowland forest sites were initially scouted and sampled; one at the Yangambi Biosphere Reserve (Tshopo province) and one in the protected forests surrounding the town of Djolu (Maringa-Lopori-Wamba Landscape, Tshuapa province). Table S1 gives detailed information on location, soil type, sampling duration, replication, periodicity, and the total number of fluxes measured at each site.

**$N_2O$ and $CH_4$ soil fluxes and $N_2O$ isotopic composition.** Soil $N_2O$ and $CH_4$ fluxes were measured using the non-steady-state, static chamber method[24] with chamber dimensions of $d = 30 \, cm$ and $h = 30 \, cm$. For the majority of the measurement periods five polyvinylchloride chambers, equipped with vent tube, thermocouple, and sampling port, were permanently installed at each site from which flasks were sampled once a week or once every two weeks and during the intensive campaigns at least once per day. Within the entire project time frame, chambers were occasionally relocated within the sites, with the core lowland site (Yoko Forest Reserve) having 16 different chamber locations, the core montane site (Kahuzi-Biéga National Park) 12 different chamber locations, and the core swamp forest site (Jardin Botanique d'Eala) 6 different chamber locations. Those chambers

affected by seasonal flooding at the swamp forest site were measured for as long as possible and replaced by floating chamber ($V = 17$ L) measurements during complete inundation. During the wet season, boardwalks were installed to sample forest water away from the flood water edge whereas during the dry season, the receding water was followed and the floating chamber operated from the edge. We separated data between chamber measurements taken under visibly inundated and non-inundated conditions. To measure the gas flux, chambers were closed over the duration of one hour and headspace air samples were taken at four evenly spread time points ($t_1 = 0$, $t_2 = 20$, $t_3 = 40$, $t_4 = 60$ min) using a disposable airtight plastic syringe which was connected to the chamber via a three-way Luer-lock valve, connecting syringe, chamber, and needle. After connecting the syringe to the sampling port, the plunger was moved a couple of times to ensure air mixing inside the chamber before actual sampling. At each time interval pre-evacuated exetainers (12 mL; Labco, UK), sealed with an additional silicon layer (Dow Corning 734, Dow Silicones Corporation, USA), were filled with 20 mL headspace air. The resulting overpressure prevents air ingress due to temperature and pressure changes potentially occurring during transport. The obtained relationship between time and concentration increase for each chamber measurement was then used to infer the individual flux rates using the ideal gas law $PV = nRT$:

$$n = \frac{PV}{RT} \qquad (1)$$

$$F = \frac{\Delta n}{\Delta t} S^{-1} \qquad (2)$$

with $n$ moles of gas [mol], $P$ partial pressure of trace gas [atm µmol mol$^{-1}$], $V$ chamber volume [L], $R$ gas constant 0.08206 [L atm K$^{-1}$ mol$^{-1}$], $T$ headspace temperature [K], $F$ flux of gas [µmol m$^{-2}$ s$^{-1}$], $\frac{\Delta n}{\Delta t}$ rate of change in concentration [mol s$^{-1}$], and $S$ surface area enclosed by chamber [m$^2$]. Headspace temperature was measured at each sampling interval using thermocouples attached to each chamber (Type T, Omega Engineering, Stamford, USA). The coefficient of determination ($R^2$) of the linear regression of $\frac{\Delta n}{\Delta t}$ was higher than 0.95 for 76% of the N$_2$O data and 75% of the CH$_4$ data (Fig. S7). Note, low $R^2$ values were mostly associated with low or zero fluxes. Thus, to not bias the data towards high fluxes, none of the data has been discarded from the analysis. Signs follow the micrometeorological convention with negative numbers denoting a flux from the atmosphere to the biosphere. All gas samples were shipped to Switzerland and analyzed at ETH Zurich for N$_2$O and CH$_4$ mole fraction using gas chromatography (456-GC, Scion Instruments, UK) with a suite of standards covering the expected range of concentrations.

For the majority of flux measurements, volumetric water content (ECH$_2$O-EC5, Meter Environment, Pullman, USA) and soil temperature (thermocouple Type T, Omega Engineering, Stamford, USA) data from 30 cm depth were available (Fig. S2). Volumetric water content was converted to WFPS using soil bulk density.

Moreover, the intramolecular distribution of $^{15}$N within the linear N$_2$O molecule ($^{15}$N site preference) was analyzed, together with the bulk $^{15}$N content ($\delta^{15}$N) and $^{18}$O content ($\delta^{18}$O) of N$_2$O, to assess whether N$_2$O production was dominated by either nitrification or denitrification. To determine $^{15}$N site preference, $\delta^{15}$N and $\delta^{18}$O of surface flux N$_2$O additional samples were taken during the high-frequency sampling campaigns at all sites (Table S1) and the data was analyzed according to a two-source mixing model approach as described in Krüger et al. (2001)[70]. In brief, two additional 180 mL gas samples were collected during gas flux measurements with a 60 mL syringe immediately after chamber closure ($t_1 = 0$ min) and shortly before chamber opening ($t_4 = 60$ min). The gas samples were then stored in pre-evacuated 110 mL serum crimp vials and transported back to Zurich, where the isotopic composition of N$_2$O was measured using a gas preparation unit (Trace Gas, Elementar, Manchester, UK) coupled to an isotope ratio mass spectrometer (IRMS, IsoPrime100, Elementar, Manchester, UK). The gas preparation unit was modified with an additional chemical trap (0.5 inches diameter stainless steel), located immediately downstream from the autosampler to pre-condition samples. This pre-trap was filled with NaOH, Mg(ClO$_4$)$_2$, and activated carbon in the direction of flow and was designed as a first step to scrub CO$_2$, H$_2$O, CO, and VOCs, which otherwise would cause mass interference during measurements. All $^{15}$N/$^{14}$N and $^{18}$O/$^{16}$O sample ratios are reported relative to AIR-N$_2$ and VSMOW, respectively, using the δ notation. For a detailed description of the analytical procedure see Verhoeven et al. (2019)[71]. Since the accuracy of $\delta^{15}$N, and $\delta^{18}$O resulting from the mixing model approach highly depends on the concentration difference between start and end of the measurement, data with $\Delta$N$_2$O < 31 ppb was discarded from further analysis[72].

**N$_2$O and CH$_4$ concentrations and N$_2$O isotopic composition dissolved in headwater streams**. During one hydrological year samples of headwater streams, which drained the catchments of the core sites for soil–atmosphere flux measurements, were taken fortnightly using the headspace equilibration technique (03/2018–03/2019 montane, lowland; 11/2019–11/2020 swamp). Further, to assess spatial variability, several different headwater streams from other lowland and montane catchments close to the core sites were additionally sampled using also the headspace equilibration technique—once in the wet season (03/2018) and once in the dry season (09/2018) (Table S1, Figs. S5 and S6). Water samples were taken in the middle of the stream using a 12 mL plastic syringe and 6 mL of air bubble-

free water injected into N$_2$-pre-flushed Labco exetainers. After the initial injection of 3 mL, a vent needle was inserted, and the remaining 3 mL of water was filled into the vial. Prior to capping and flushing with ultrapure N$_2$ (Alphagaz II, Carbagas, Switzerland), the 12 mL Labco exetainer was conditioned with 50 µL of 50% (w/v) ZnCl$_2$ to suppress microbial activity after sample injection[73]. Samples were transported back to Zurich and headspace was analyzed for N$_2$O and CH$_4$. Final dissolved gas concentrations were calculated according to Henry's law with a detailed protocol provided in the supplement.

Similar to the preparation of the Labco exetainers, 110 mL serum crimp vials used for the isotopic characterization of dissolved N$_2$O were pre-flushed with ultrapure N$_2$ and pre-treated with 200 µL of 50% (w/v) ZnCl$_2$ to stop microbial activity after sample injection. Once a month, during the same year as dissolved gases were taken, 20 mL of riverine water was injected into the pre-flushed 110 mL serum vials (only montane and lowland streams). Back in the lab, vial headspace was analyzed for N$_2$O isotopic composition using the same IRMS as described above. 20 mL water in 110 mL serum crimp vials allowed direct headspace measurements without the needle of the autosampler entering the water when injected into the vial. The IRMS system operates with a double-holed needle, flushing the entire sample from the headspace using helium as carrier gas. Since samples were measured in the actual gas phase, $\delta^{18}$O and $\delta^{15}$N values of the dissolved N$_2$O phase were calculated using the equilibrium fractionation factors given in Thuss et al. (2014)[43]:

$$\alpha_{eq} = \frac{R_{headspace}}{R_{dissolved}}$$

with $\alpha_{eq} = 0.99925$ for $^{15}$N and 0.99894 for $^{18}$O and $R_{headspace} = (\delta_{headspace}/1000 + 1) \cdot R_{reference}$. Corresponding reference values (VSMOW for $\delta^{18}$O; AIR-N$_2$ for $\delta^{15}$N) were taken from Werner ad Brand (2001)[74].

**Statistics**. We used the long-term datasets from Kahuzi-Biéga National Park, Yoko Forest Reserve, and Jardin Botanique d'Eala to assess intra-site (between chamber) and intra-annual variability in the N$_2$O and CH$_4$ fluxes. For between chamber variance, we calculated the arithmetic average of the fluxes per chamber over the monitoring periods (see Supplementary Table S2) and then calculated the variance, and subsequently also the standard deviation and coefficient of variance over those 'chamber averages' per site. For the intra-annual variability, an arithmetic average was calculated per week per site (using the different chamber flux estimates). Subsequently, the variance over those weekly site fluxes was calculated, along with the standard deviation and coefficient of variation per site.

We estimated median fluxes (from a highly positively skewed distribution of fluxes) for both N$_2$O and CH$_4$. This was done by fitting linear mixed effect models per forest type (lowland, montane, and swamp forest) with log-transformed fluxes as a response variable, without fixed effects, and with chamber ID (nested within the plot) and sampling date as a random intercept. The effect estimates for the intercept of the different models, corroborating with a 'best estimator' for the fluxes in the different forest types, were exponentiated for the estimation of median fluxes per forest type. Finally, we fitted another set of linear mixed effect models to quantify the effects of recorded field temperature and water-filled pore space on the fluxes. For this, different models were fitted for the three different forest types (lowland, montane, and swamp). After log-transforming the response variables, i.e. the fluxes of, respectively, N$_2$O and CH$_4$, linear mixed effect models were fitted using temperature and water-filled pore space as fixed effects, and chamber ID as a random intercept. The estimated fixed effects were extracted from the models and recalculated to the % change that both individual effects cause in the response data by (exp(effect)−1)*100. Note, for visualization purposes, collected soil and riverine data were pooled into weekly classes irrespective of year.

All statistical analyses were carried out using the R software[75], mixed effect models were fitted using the lme4 package[76], and $R^2$ values for mixed effect models were calculated following Nakagawa and Schielzeth (2013)[77].

## Data availability
The core datasets generated during the current study have been deposited in the Zenodo repository [https://zenodo.org/record/5764853#.Ya-jJIPTXOR] and are also available from the corresponding author upon request.

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

## Acknowledgements

The authors thank the park guards of the *Institute Congolais pour la Conservation de la Nature* (ICCN) for support during field campaigns. We further acknowledge the support by the *Institut National des Etudes et Recherches Agronomique* (INERA) during our fieldwork in the Yangambi Biosphere Reserve, and the African Wildlife Foundation (AWF) during our work in the Maringa-Lopori-Wamba landscape. M. Barthel was funded by two Swiss National Foundation exchange grants (IZSEZ0_186376; IZK0Z2_170806). In addition, M. Barthel, T.W.D., and J.S were supported through ETH core funding.

## Author contributions

M. Barthel and J.S. conceived the study. M. Barthel, M. Bauters, S.B., N.G., N.D., L.S., T.W.D., I.A.M., N.M.B., C.E.M., D.M.M., and J.K.M. conducted the fieldwork. J.M and B.W. provided reference gases for N$_2$O isotope analysis. C.B., F.M.M., L.C.N., G.L.E., R.G.M.S., P.B., G.B., J.Z.M., B.V., M.B.R., K.V.O., and J.S. facilitated local collaborations. M. Bauters did the statistical data analysis. T.W.D., M. Bauters, S.B., and J.S. helped M. Barthel with the drafting of the manuscript outline. All co-authors substantively revised the manuscript.

## Competing interests

The authors declare no competing interests.
