## [Peer Review File · Nature Communications]

Low N₂O and variable CH₄ fluxes from tropical forest soils of the Congo BasinReviewers' Comments:

Reviewer #1:

Remarks to the Author:

This long-term and rigorous research generated invaluable ground-based data of soil N₂O and CH₄ fluxes for forests of the Congo basin. It also investigated in depth hypothesis and mechanisms underlying the magnitude of the fluxes. I enjoyed very much reading this paper which is extremely well written and appreciate the high quality of the work. I congratulate the authors for their research and their manuscript.

Below are some small comments and suggestions for improving the manuscript.

Abstract

I. 44-46 "Here, we fill this knowledge gap with data derived from 1267 individual on-ground N₂O and CH₄ flux measurements together with 158 riverine dissolved concentrations."

Could authors mention instead the number of sites monitored, over which period, and at which frequency? A simple number of flux measurement is not very helpful.

I. 48 "from Congo lowland forests."

Authors: Please, specify you are referring to upland / unflooded forests. The Congo basin is also home to a large area of wetland forests which are expected to be a large CH₄ source.

I. 50 "as current estimates suggest"

Where do these estimates come from if flux measurements are scarce? Top-down estimates?

I. 51 "a more nuanced accounting"

Unclear: Nuanced in which sense?

I. 59-104 The Central African Data Gap

Please, clearly indicate to the reader that this paper refers to upland forests.

I.65-66 "high N₂O emissions are mostly attributed to excess nitrogen relative to phosphorus in these soils"

Are findings from the Amazon region generalizable to the whole tropics?

I. 68 "suggested, through modelling"

Which type of modelling? Process-based modelling?

I. 68 "emission profile inferred from similar moist tropical forests"

Please provide more detailed information. The reader who does not know the corresponding literature doesn't clearly understand how such extrapolations were made.

I. 81 "supposedly"

Not sure this term is truly needed. As authors argued beforehand the studied area is large and is therefore expected to contribute importantly to non-CO₂ emissions / uptakes.

I. 94 "1267 individual flux measurements are available for this study (Fig 1; Table S1)"

In Table S1, please, provide length of experiment and sampling frequency for terrestrial / aquatic measurements per location.

I. 107 "For each forest type, N₂O and CH₄ fluxes were stable and consistent within and across sites (Figure 2)." and I. 111- 112 "the weekly time resolution provides a representative interval for estimating annual budgets"

In Figure 2, the 2016 - 2019 years were pooled into weekly classes. Did locations display inter-annual variability in fluxes?

Figure 2 "Note, for visualization purposes extreme values with $>3 \text{ nmol m}^{-1} \text{ s}^{-1}$ were not shown ($n = 7$)"

It would be interesting to know the magnitude of these fluxes. Could authors, please, add high flux values in SI?

Figure 2 Please, provide n value for montane and lowland average annual fluxes independently.

Fig S1 This Figure is confusing and gives the impression that all locations were not evenly sampled with:

- Monthly measurements conducted at one location only and over a limited period of 3 months;
- Daily measurements conducted at three sites over one period of about 10-15 days at two locations, and two periods of about 10-15 days at another location;
- Sub-daily measurements conducted once at two locations.

I may be misreading this Figure but I think differences in sampling frequency / regimes among locations should be more clearly stated at the beginning of the manuscript. The legend of this Figure should be clarified.

I. 120-121 "low variability of soil temperature and WFPS within a given location (data not shown)." Please, display the data in supplementary information as well as annual averages.

I. 122-123 "releasing on average $0.59 \text{ kg N}_2\text{O-N ha}^{-1} \text{ yr}^{-1}$ from montane and $1.38 \text{ kg N}_2\text{O-N ha}^{-1} \text{ yr}^{-1}$ from lowland forests." Please, provide standard error for all mean annual fluxes.

I. 123-124 "Our lowland tropical forest values -representing 90% of Congo Basins dense forest" Please, provide a reference to support this statement.

I. 126-130 "We estimate, using the vegetation cover maps of Verhegghen et al. (2012)²⁸ and recently published numbers¹⁴, that only 4.2% (instead of 5.9%) of yearly global N_2O released from soils under natural vegetation is contributed by the Congo Basin's dense humid lowland forests with an absolute total of $0.233 \text{ Tg N}_2\text{O-N yr}^{-1}$ corresponding to $0.1 \text{ Tg N}_2\text{O-N yr}^{-1}$ less emitted N_2O ." Could authors, please, provide, details of these computations as supplementary information?

I. 136 "Some erratic and rare extremely high CH_4 emission fluxes were observed" Please, provide the magnitude of those fluxes.

I. 279-281 "with the long term lowland site (Yoko) having 16 different chamber locations and the montane site (Kahuzi- Biéga) 12 different chamber locations." How many chambers were deployed at the Yangambi Biosphere Reserve and in the protected forests surrounding the town of Djolu?

I. 301 "All gas samples were shipped to Switzerland and analyzed at ETH Zurich" How long after sampling were the concentrations analyzed? Were the exetainers sealed?

Reviewer #2:

Remarks to the Author:

In this manuscript, the authors report soil-atmosphere fluxes of emissions of N_2O and CH_4 from soils at four sites (3 lowland, 1 montane) in the Congo Basin using a closed-chamber method; they augment this dataset with dissolved N_2O and CH_4 concentrations in headwater river samples and 15N data from the soil flux measurements to determine whether N_2O emissions were driven primarily by nitrification or denitrification. Finally, the authors use the collected soil flux measurements to revise

the current estimates of CH₄ sequestration rates and N₂O emissions rates at the scale of the Congo Basin, reporting that CH₄ sequestration in the Basin is being underestimated and N₂O flux from the Basin is being overestimated (by ~30% and ~70%, respectively).

This study contributes a robust set of soil greenhouse gas (GHG) flux data from a region and set of ecosystems that are chronically under-sampled and under-represented in global databases of soil GHG flux – as the authors note in Figure 1, there is nearly no soil N₂O and CH₄ flux data available for this entire area of a continent, so on that score alone this contribution is very valuable. The addition of isotopic work to get at what processes are driving N₂O emissions is also a novel contribution that is not only novel within the Congo Basin, but which adds to the very small number of similar studies across the tropics more broadly.

My primary piece of constructive feedback about this paper is the relative lack of explanation around the regional extrapolation exercise, which yields some of the paper's most emphasized findings. Specifically, I refer to Line 126: As far as I can tell, how the authors conducted this global extrapolation is not explained in the methods. Please include a short summary of this approach, particularly since this extrapolation plays a key role in the results highlighted in the abstract. Was this a simple extrapolation in which the flux numbers from all sites were projected over the forest coverage zones in Verhegghen et al. 2012? Verhegghen et al. 2012 does report on submontane/mountain forest vs. lowland forest, I believe – were the flux estimates from those sites in this dataset scaled proportionally based on the Verhegghen et al. 2012 biomes distribution or similar? Or were the lowland and montane results here combined? Was there a sensitivity analysis conducted or similar uncertainty exercises included? E.g., the projected total N₂O emissions reported as 0.233 Tg N₂O per year (line 130) doesn't include a 95% CI or similar. Is a range of estimates reported elsewhere that I perhaps missed? Along the same lines, there is no analogous "Basin-total" sequestration extrapolation reported for CH₄ (lines 134-146).

I believe that this manuscript would be greatly strengthened by either reframing away from such a strong emphasis on these extrapolation exercises, or by making the calculation of these extrapolations must more robust, and more explicitly justified and detailed in the methods.

Finally, hats off to the authors for collecting a dataset with very solid temporal coverage under what must have been a challenging set of logistics/field conditions. These data substantially improve our understanding of soil GHG flux patterns in the Congo Basin.

SPECIFIC COMMENTS

- Citation 22: Still marked as "in revision" – typo?
- Fig. 2: It would be more informative to include smoothed conditional mean lines for lowland vs. montane sites, since you note in line 107 that fluxes were stable/consistent across sites "for each forest type" – ought the reader be able to more clearly distinguish between the forest types, then?
- Line 120: Might be worth including these data as a supplemental table with simple summary stats, including a measure of variation across the sample.
- Figure S5 and methods line 299-300: The authors should justify why fluxes with very low R² values were not discarded. Though 80/77% of the N₂O and CH₄ fluxes, respectively, had good linear fits, Figure S5 indicates that a small proportion of the fluxes have exceedingly poor fits (R² < 0.1, for instance). E.g., Were these determined to be fluxes close to zero (which are often discarded due to poor flux fits), which weren't discarded in order to not bias the sample towards high fluxes? Is there some other systematic explanation for retaining fluxes in the sample with a poor model fit? The fact that many of the fluxes have a solid fit doesn't on its face justify keeping samples with a poor model

fit. Please explain/justify.

- It would strengthen the paper to briefly compare (bearing in mind the tight word limit at play, of course) the Fig. 1 data coverage with geographic trends in soil GHG data from other regions (see recommended citations for a big-data approach to this angle).
- Recommended citation: Bond-Lamberty, B., Christianson, D. S., Malhotra, A., Pennington, S. C., Sihi, D., AghaKouchak, A., ... & Zou, J. (2020). COSORE: A community database for continuous soil respiration and other soil-atmosphere greenhouse gas flux data. *Global change biology*, 26(12), 7268-7283.
- Recommended citation: Dorich, C. D., Conant, R. T., Albanito, F., Butterbach-Bahl, K., Grace, P., Scheer, C., ... & van der Weerden, T. J. (2020). Improving N₂O emission estimates with the global N₂O database. *Current Opinion in Environmental Sustainability*, 47, 13-20.

Reviewer #3:

Remarks to the Author:

This paper is very important as the authors report measurements of nitrous oxide and methane fluxes in a very understudied region of the world: tropical rainforests of the Congo Basin. They find that these forests store more methane and release less nitrous oxide than previously reported or understood. The overall goals of the paper are strong, but the text related to the Methods make it very difficult to assess the validity of the authors' conclusions and the conclusions they draw are too grand based on their sampling design. Here are my main concerns with the paper with detailed comments below:

1. If I understand correctly, the authors had a total of five locations per site (as mentioned in line 276) to measure methane and nitrous oxide fluxes, which is too small to make such large conclusions about tropical forests of the Congo Basin as a whole. Instead, I recommend they limit their conclusions to the sites they measured. Soil trace gas fluxes are notoriously heterogeneous, making a sample size of $n = 5$ far too small to justify the grand conclusions made in this paper.
2. The study design is very unclear as written, making it difficult to assess the validity of their results. For example, why is there one montane, but three lowland sites included? What distinguishes the long-term from short-term sampling? The authors explain the number of static chambers ($n=5$ per site), but not how apart they are from each other, whether all factors were kept constant across them within a site, or whether they were chosen randomly to capture the heterogeneity of each of the four sites.
3. The authors do not provide the dates of sampling for their soil gas measurements. Knowing whether they overlapped in time with the riverine sampling is essential for determining whether their conclusions comparing soil and riverine data analyses and making conclusions about land to water fluxes.
4. The sample size the authors present in Table S1 is misleading because it provides the total number of times the authors measured fluxes, not the total number of replicates they have per site (which I believe is only 5).

Specific suggestions:

Lines 42 and 43: Define methane and nitrous oxide before abbreviating

Lines 45-46: A much better set of numbers to provide here is the total number of replicate chambers per site (which I believe is only 5 per site).

Figure 1: This figure is confusing as currently shown. The Abstract states that the authors compiled

data from 1267 individual on-ground measurements and 158 riverine dissolved concentrations across the Congo Basin, yet this map shows four sites total in the Congo Basin and it's unclear how the measurements are distributed between on-ground vs. riverine measurements, nor between montane and lowland sites. The authors need to make this map much more informative than what is currently shown.

Lines 73-76: The authors bring up the great points of how tropical forests in the Congo Basin are different from forests elsewhere. I suggest they interpret their results in this context, providing much more detail about their sites and how their soils, plants, and other factors may have influenced their results.

Line 84: Replace "none date" with "no data"

Lines 83-84: Does this one paper correspond to the four sites shown on the map inside the Congo Basin in Figure 1? If not, does it overlap with these four sites?

Lines 89-90: Here the text sounds like there was just one lowland and one montane forest site. Further down, the authors state that there were measurements taken at "two additional sites" (lines 93-94), but they don't designate whether these are lowland or montane sites. I had to go down to the Figure 2 legend to understand that there is just one montane forest site (Kahuzi-Biega National Park) and three lowland forest sites. I suggest that the authors make it much clearer earlier in the text (I suggest in lines 89-91) their sites and explain why they had just one montane site, but three lowland sites. Why not an equal distribution of samples? Why not 2 lowland and two montane?

Figure 2: I suggest splitting up the smoothed line to separate the one montane from the three lowland sites. I also suggest adding the actual dates to the X-axis so readers better understand the timing of the study.

Line 107: The authors state that "For each forest type, N₂O and CH₄ fluxes were stable and consistent within and across sites", but the way the data are presented in Figure 2 makes it impossible for the reader to decide that for themselves. I suggest, as mentioned above, that the authors present the results of their statistical analyses separately for the four sites. Or at a minimum separate them by the 1 montane vs. 3 lowland forest sites. If the authors are actually referring to the results shown in Figure S1, they should refer to it here.

Lines 122-123: The authors state "Soil N₂O fluxes ranged from -0.108 to 6.42 nmol m⁻² s⁻¹... releasing on average 0.59 kg N₂O-N ha⁻¹ yr⁻¹ from montane and 1.38 kg N₂O-N ha⁻¹ yr⁻¹ from lowland forests." Again, it's really hard for the reader to see these differences between sites since the data are not presented separately by forest site.

Lines 138-139: Same comments for methane as mentioned above for nitrous oxide.

Lines 201-214: The authors could describe their forests in much more detail here to try to explain why the results they found are distinct from other tropical forest ecosystems around the globe.

Lines 256-271: The methods text is missing a lot of essential information. The paragraph needs a short description of what distinguishes the "long-term observation sites" vs. "short sampling campaigns". Also, the replication within each of the four sites is very unclear (is it in fact n=5 chambers per site, so n=20 total?). When were the soil gas flux samples taken? The authors explain the dates for stream sampling (March 2018 through February 2019 in line 326), but no such dates are provided for the soil gas measurements (though I see "2016-2019 in the legend for Fig S4). The authors state on line 91 that samples were collected over three years, which three? For how many weeks total? This should be much clearer in the main text of the paper.

Line 305: "To determine site preference" is unclear. The authors need to explain this.

Line 328-329: The authors state that "To assess spatial variability, several different headwater streams from other catchments close to the flux chamber sites were also sampled.." but they provide no details. They need to include the number of sites total, how far they were from the soil sampling sites, how samples were taken, how statistical analyses were conducted, etc.,

Line 357: The authors state "For the decomposition of variance, we used the complete lowland and montane dataset." Does this mean they have one montane and three lowland sites in this particular analysis? Were all four sites treated the same way? How is that justified?

Lines 374-375: The authors note that "soil and riverine data collected were pooled into weekly classes irrespective of year only for visualization purposes", but they didn't provide the years for when soils were sampled. Were the soils sampled at the same time as river samples?

Figure S1, S2, S4: I suggest adding the actual years to the X-axis of all three figures so readers better understand the timing of the study.

Figure S3: A map with these sites would strengthen the paper. At a minimum, the distance between these sites, the dominant tree species, soil types, etc., are needed to better understand how well these sites represent the region.

Figure S3: What is the sample size for each of these sites? It is difficult to assess these results without that. I don't mean the number of times they measured fluxes, I mean the number of spatial replicates utilized.

REVIEWER COMMENTS

Reviewer #1 (Remarks to the Author):

This long-term and rigorous research generated invaluable ground-based data of soil N₂O and CH₄ fluxes for forests of the Congo basin. It also investigated in depth hypothesis and mechanisms underlying the magnitude of the fluxes. I enjoyed very much reading this paper which is extremely well written and appreciate the high quality of the work. I congratulate the authors for their research and their manuscript.

Below are some small comments and suggestions for improving the manuscript.

We thank the reviewer for this very positive feedback on our work. This is very encouraging! We would like to mention here that we now have added data from a swamp forest to our current dataset. This new data was obtained from samples which we retrieved and analyzed during the first revision of the paper and we think it is a great complement to the current dataset as we now cover all major forest types of the Congo Basin: montane, lowland, and swamp forest. Please find below a point-by-point response to the comments and suggestions given, taking into account the newly added site.

Abstract

l. 44-46 “Here, we fill this knowledge gap with data derived from 1267 individual on-ground N₂O and CH₄ flux measurements together with 158 riverine dissolved concentrations.”

Could authors mention instead the number of sites monitored, over which period, and at which frequency? A simple number of flux measurement is not very helpful.

We agree with the reviewer that a more detailed information on the spatial and temporal coverage is needed. We kept, however, the simple numbers of flux measurements, as we would like to emphasize that we have a high-resolution dataset from which we drew our conclusions. We re-wrote the abstract and added the following information:

“Here, we provide new multi-year data derived from on-ground soil flux ($n = 1558$) and riverine dissolved gas concentration ($n = 332$) measurements spanning montane, swamp, and lowland forests. Each forest type core monitoring site was sampled at least for one hydrological year between 2016 - 2020 at a frequency of 7-14 days.”

l. 48 “from Congo lowland forests.”

Authors: Please, specify you are referring to upland / unflooded forests. The Congo basin is also home to a large area of wetland forests which are expected to be a large CH₄ source.

The reviewer is absolutely correct by pointing out this important detail. Originally, we were referring to an ‘upland’ lowland forest type which is not seasonally flooded. Since we have now added a swamp forest site to the manuscript, this distinction should be clear to the readership.

l. 50 “as current estimates suggest”

Where do these estimates come from if flux measurements are scarce? Top-down estimates? We compared our numbers to studies based on meta-analysis or modelling efforts. We have, however, deleted this sentence from the abstract to be in line with the current changes of the manuscript.

l. 51 “a more nuanced accounting”

Unclear: Nuanced in which sense?

We agree that this statement is a bit vague. We meant nuanced in the sense of ‘detailed’. We now removed this sentence entirely from the abstract for word limit reasons and added a rephrased statement to the end of the manuscript instead. It reads now:

“Our study is the first step towards addressing this uncertainty and highlights both the variation and nuanced nature of these key biogeochemical fluxes.”

l. 59-104 The Central African Data Gap

Please, clearly indicate to the reader that this paper refers to upland forests.

The paragraph starting from line 88 is now modified taking into account the newly implemented swamp forest flux data. It now reads:

“To achieve this, we quantified seasonal soil-atmosphere fluxes of N₂O and CH₄ from the three major tropical forest types (montane, lowland, swamp) within the Congo Basin at a weekly to fortnightly resolution using the static chamber method (Hutchinson & Mosier, 1981). Initial short-term campaigns with daily measurements were conducted in 2016 and 2017 at three lowland locations (Maringa-Lopori-Wamba Landscape, Yangambi Biosphere Reserve, Yoko Forest Reserve) and one montane location (Kahuzi-Biéga National Park) to constrain spatiotemporal variation (Fig. S1). Expanding on these short-term expeditions, long-term observation ‘core sites’ were established in Yoko Forest Reserve and Kahuzi-Biéga National Park, as lowland and montane representatives, starting in 2016 and 2017, respectively. These two long-term core sites were subsequently expanded by the addition of a seasonally inundated swamp forest core site in 2019 (Jardin Botanique d’Eala). Including the initial short-term campaigns, 1558 individual soil-atmosphere flux measurements were undertaken for this study (Table S1).”

l.65-66 “high N₂O emissions are mostly attributed to excess nitrogen relative to phosphorus in these soils”

Are findings from the Amazon region generalizable to the whole tropics?

While we think that findings from the Amazon cannot be generalized to the whole tropics because of ecological, hydrological, and pedological differences we do hope to identify generalizable mechanisms which are at play throughout the tropics to be able to better model and predict tropical biogeochemical cycles across the tropics. We refer to this Amazon study as we do currently do not have any knowledge about the Congo Basin at hand.

l. 68 “suggested, through modelling”

Which type of modelling? Process-based modelling?

It was suggested through process-based modelling. We have added this information.

l. 68 “emission profile inferred from similar moist tropical forests”

Please provide more detailed information. The reader who does not know the corresponding literature doesn’t clearly understand how such extrapolations were made.

We agree that this is a bit of a too vague and generalized statement and therefore removed this side sentence from the paper.

l. 81 “supposedly”

Not sure this term is truly needed. As authors argued beforehand the studied area is large and is therefore expected to contribute importantly to non-CO2 emissions / uptakes.

We would like to keep this term because until now the Congo Basin was just assumed to be a global emission/uptake hotspot, but evidence is still lacking.

l. 94 “1267 individual flux measurements are available for this study (Fig 1; Table S1)”
In Table S1, please, provide length of experiment and sampling frequency for terrestrial / aquatic measurements per location.

This is an excellent idea. Please see now the amended Table S1 and rephrased corresponding paragraph:

Table S1 | General site information, including sampling duration and periodicity of different datasets used. Core long-term observation sites are highlighted in bold. Terrestrial and aquatic sites are marked in grey and blue respectively.

LOCATION Latitude, longitude, elevation	FOREST/ SOIL TYPE	DATES	PERIODICITY	REP	n	n isotopes		
Kahuzi-Biéga National Park S 2.31314, E 28.75471, 2050 m	Montane Umbric Ferralsol	06.09.2016 - 19.09.2016 [§] 26.04.2017 - 02.05.2017 [§] 06.05.2017 - 24.03.2018 29.03.2018 - 26.03.2019*	1 d 1 d 7 d 14 d	7 3 3 5	} 398	12		
Jardin Botanique d'Eala N 0.06335, E 18.31054, 300 m	Swamp Eutric Gleysol	19.11.2019 - 25.11.2019 26.11.2019 - 07.12.2020	1 d 14 d	5 6			} 266	6
Yoko Forest Reserve N 0.29299, E 25.30111, 486 m	Lowland Xanthic Ferralsol	26.09.2016 - 09.10.2016 [§] 05.05.2017 - 01.06.2017 [§] 31.10.2016 - 18.12.2017 17.03.2018 - 30.03.2019* 22.04.2019 - 01.03.2020	1 d 1 d 7 d 14 d 14 d	7 6 6 5 5				
Yangambi Biosphere Reserve N 0.81128, E 24.49041, 480 m	Lowland Xanthic Ferralsol	11.10.2016 - 24.10.2016	1 d	7			98	7
Maringa-Lopori-Wamba Landscape N 0.67208, E 22.33583, 386 m	Lowland Xanthic Ferralsol	27.04.2016 - 01.08.2016	irregular	3	42	6		
TOTAL TERRESTRIAL MEASUREMENTS					1558	63		
Langa River S 2.30674, E 28.75554	Montane Umbric Ferralsol	29.03.2018 - 26.03.2019	14 d	3	80	14		
Yoko River N 0.29320, E 25.29481	Lowland Xanthic Ferralsol	17.03.2018 - 30.03.2019	14 d	3	78	9		
Lolifa River S 0.03135, E 18.3102	Swamp Eutric Gleysol	26.11.2019 - 07.12.2020	14 d	3	84			
Buba River S 2.33138, E 28.75651	Montane Umbric Ferralsol	29.03.2018, 27.08.2018		5	10			
Cirado River S 2.333778, E 28.75445	Montane Umbric Ferralsol	29.03.2018, 27.08.2018		5	10			
Isalowe River N 0.79808, E 24.50101	Lowland Xanthic Ferralsol	13.03.2018, 19.09.2018		5	10			
Boonde River N 0.77434, E 24.39577	Lowland Xanthic Ferralsol	13.03.2018, 19.09.2018		5	10			
Lobilo River N 0.76081, E 24.59414	Lowland Xanthic Ferralsol	15.03.2018, 19.09.2018		5	10			
Lokombe River N 0.71624, E 24.605	Lowland Xanthic Ferralsol	15.03.2018, 19.09.2018		5	10			
Bekele River N 0.69606, E 24.60126	Lowland Xanthic Ferralsol	15.03.2018, 19.09.2018		5	10			
Losa River N 0.6587, E 24.64186	Lowland Xanthic Ferralsol	15.03.2018, 19.09.2018		5	10			
Lusambila River N 0.75728, E 24.48266	Lowland Xanthic Ferralsol	13.03.2018, 19.09.2018		5	10			
TOTAL AQUATIC MEASUREMENTS					332	23		

[§]N₂O dataset partly published in Bauters et al. 2019; *N₂O dataset published in Gallarotti et al. 2021.

~ Period during which complementing riverine samples were taken from headwater streams draining the same catchments in which core sites were located.

l. 107 “For each forest type, N₂O and CH₄ fluxes were stable and consistent within and across sites (Figure 2).” and l. 111- 112 “the weekly time resolution provides a representative interval for estimating annual budgets”

In Figure 2, the 2016 - 2019 years were pooled into weekly classes. Did locations display inter-annual variability in fluxes?

We did not investigate inter-annual variability as we have only one site at the lowland which covers a period of 3 full years (2016-2019). However, since we did not measure at the same frequency during these 3 years and had small gaps, such inter-annual comparison is methodological challenging as one will have to gap-fill missing fluxes. Such flux gap-filling is commonly done for CO₂ in the eddy-covariance realm but difficult to do for N₂O and CH₄. However, we have reworked this entire section and redone the statistics given that we have implemented a new site. We now write:

“For both CH₄ and N₂O, there was considerable spatial (between chambers within sites; Table S2) and temporal (within site, across weeks; Table S2) variance, as is common for both gas species.”

Figure 2 “Note, for visualization purposes extreme values with >3 nmol m⁻¹ s⁻¹ were not shown (n = 7)”

It would be interesting to know the magnitude of these fluxes. Could authors, please, add high flux values in SI?

Figure 2 has completely reworked with y-axis dimensions now covering the full flux magnitude.

Figure 2 Please, provide n value for montane and lowland average annual fluxes independently.

Figure 2 is now separated into 3 different panels; one for montane, lowland, and swamp forest detailing respective annual fluxes and n values separately.

Fig S1 This Figure is confusing and gives the impression that all locations were not evenly sampled with:

- Monthly measurements conducted at one location only and over a limited period of 3 months;
- Daily measurements conducted at three sites over one period of about 10-15 days at two locations, and two periods of about 10-15 days at another location;
- Sub-daily measurements conducted once at two locations.

I may be misreading this Figure but I think differences in sampling frequency / regimes among locations should be more clearly stated at the beginning of the manuscript. The legend of this Figure should be clarified.

We thank the reviewer for pointing this out. This figure shows the results of the initial sampling campaigns, from which we chose the permanent core sites. We have changed the legend and included more detailed information in the figure caption. We have also amended the text in the intro for clarity and additionally reworked Table S1.

l. 120-121 “low variability of soil temperature and WFPS within a given location (data not shown).”

Please, display the data in supplementary information as well as annual averages.

Figures of soil temperature and soil moisture are now added to the supplementary information (Fig S2a, b), with both figures including specific forest type averages.

l. 122-123 “releasing on average 0.59 kg N₂O-N ha⁻¹ yr⁻¹ from montane and 1.38 kg N₂O-N ha⁻¹ yr⁻¹ from lowland forests.”

Please, provide standard error for all mean annual fluxes.

We provide now upper and lower 95% confidence intervals for all values including density distribution functions in the supplement (figures S3 and S4).

l. 123-124 “Our lowland tropical forest values -representing 90% of Congo Basins dense forest”

Please, provide a reference to support this statement.

Added (Verhegghen et al., 2012).

l. 126-130 “We estimate, using the vegetation cover maps of Verhegghen et al. (2012)²⁸ and recently published numbers¹⁴, that only 4.2% (instead of 5.9%) of yearly global N₂O released from soils under natural vegetation is contributed by the Congo Basin’s dense humid lowland forests with an absolute total of 0.233 Tg N₂O-N yr⁻¹ corresponding to 0.1 Tg N₂O-N yr⁻¹ less emitted N₂O.”

Could authors, please, provide, details of these computations as supplementary information?

This paragraph and all related conclusions have been removed from the manuscript. We now computed simple forest type weighted averages based on the spatial coverage of different forest types which is described as follows:

“...releasing a forest type weighted average of 8.5 kg CH₄-C ha⁻¹ yr⁻¹ to the atmosphere if we assume our sites to be representative of their respective forest types (weighted flux based on forest coverage²⁹: 90.6% lowland, 6.8% swamp, 2.6% montane).”

l. 136 “Some erratic and rare extremely high CH₄ emission fluxes were observed”

Please, provide the magnitude of those fluxes.

Information now implemented in the text:

“One erratic and high CH₄ flux was also observed for a single chamber at the lowland site (37.6 nmol m⁻² s⁻¹, not shown) which was likely...”

We have also added two new figures (figure S3 and S4) (for both, CH₄ and N₂O) to the supplement showing the density distribution functions separated by forest type, thus specifying distributional information, flux ranges and outlier details.

l. 279-281 “with the long term lowland site (Yoko) having 16 different chamber locations and the montane site (Kahuzi-Biéga) 12 different chamber locations.”

How many chambers were deployed at the Yangambi Biosphere Reserve and in the protected forests surrounding the town of Djolu?

This information has been added to Table S1.

l. 301 “All gas samples were shipped to Switzerland and analyzed at ETH Zurich”

How long after sampling were the concentrations analyzed? Were the exetainers sealed?

All exetainer and vials were additionally sealed with a silicon layer applied on top of the septa (Dow Corning 734). Sample storage was generally not longer than 6 months. We understand that this is somewhat a long storage time but given the remoteness and circumstances of working in DR Congo, a faster sample pipeline was unfortunately not possible. However, given the excellent coefficients of determination of the linear fits of the flux calculation and the replicate consistency of the dissolved gas concentrations, we are

confident of the data quality. That the exetainers were sealed additionally was already mentioned in a small side note but we now included also the manufacturer of the silicone to make this more prominent:

“At each time interval pre-evacuated exetainers (12 mL; Labco, UK), sealed with an additional silicon layer (Dow Corning 734, Dow Silicones Corporation, USA), were filled with 20 mL headspace air.”

Reviewer #2 (Remarks to the Author):

In this manuscript, the authors report soil-atmosphere fluxes of emissions of N₂O and CH₄ from soils at four sites (3 lowland, 1 montane) in the Congo Basin using a closed-chamber method; they augment this dataset with dissolved N₂O and CH₄ concentrations in headwater river samples and 15N data from the soil flux measurements to determine whether N₂O emissions were driven primarily by nitrification or denitrification. Finally, the authors use the collected soil flux measurements to revise the current estimates of CH₄ sequestration rates and N₂O emissions rates at the scale of the Congo Basin, reporting that CH₄ sequestration in the Basin is being underestimated and N₂O flux from the Basin is being overestimated (by ~30% and ~70%, respectively).

This study contributes a robust set of soil greenhouse gas (GHG) flux data from a region and set of ecosystems that are chronically under-sampled and under-represented in global databases of soil GHG flux – as the authors note in Figure 1, there is nearly no soil N₂O and CH₄ flux data available for this entire area of a continent, so on that score alone this contribution is very valuable. The addition of isotopic work to get at what processes are driving N₂O emissions is also a novel contribution that is not only novel within the Congo Basin, but which adds to the very small number of similar studies across the tropics more broadly.

My primary piece of constructive feedback about this paper is the relative lack of explanation around the regional extrapolation exercise, which yields some of the paper’s most emphasized findings. Specifically, I refer to Line 126: As far as I can tell, how the authors conducted this global extrapolation is not explained in the methods. Please include a short summary of this approach, particularly since this extrapolation plays a key role in the results highlighted in the abstract. Was this a simple extrapolation in which the flux numbers from all sites were projected over the forest coverage zones in Verhegghen et al. 2012? Verhegghen et al. 2012 does report on submontane/mountain forest vs. lowland forest, I believe – were the flux estimates from those sites in this dataset scaled proportionally based on the Verhegghen et al. 2012 biomes distribution or similar? Or were the lowland and montane results here combined? Was there a sensitivity analysis conducted or similar uncertainty exercises included? E.g., the projected total N₂O emissions reported as 0.233 Tg N₂O per year (line 130) doesn’t include a 95% CI or similar. Is a range of estimates reported elsewhere that I perhaps missed? Along the same lines, there is no analogous “Basin-total” sequestration extrapolation reported for CH₄ (lines 134-146).

I believe that this manuscript would be greatly strengthened by either reframing away from such a strong emphasis on these extrapolation exercises, or by making the calculation of these extrapolations must more robust, and more explicitly justified and detailed in the methods.

We thank the reviewer for pointing this out. This was also mentioned by the other two reviewers. We now simply report our flux values for the respective forest types and contextualize them in a wider scope. Consequently, we reframed the results and conclusions away from the extrapolation outcome and deleted the original sections elaborating on it. We give, however, a simple basin-wide average flux based on the percentage covered by the respective forest type but clearly state that:

“We would like to stress, however, to not overinterpret these results as much more spatiotemporal data coverage is needed to draw basin-wide conclusions as to whether...”

Finally, hats off to the authors for collecting a dataset with very solid temporal coverage under what must have been a challenging set of logistics/field conditions. These data substantially improve our understanding of soil GHG flux patterns in the Congo Basin.

We very much appreciate this comment. Thank you. Before we give a detailed response to the specific comments and suggestions made below, we would like to point out that we have now added data from a swamp forest to our current dataset. We were lucky to be able to retrieve these samples from the DR Congo during the pandemic and thought it is a great complement to the existing dataset as we now cover all major forest types of the Congo Basin: montane, lowland, swamp. That is, our point-by-point response now takes into account the newly added site. Despite this new addition, we maintained the original manuscript structure and only completely re-wrote the section ‘Fluxes of N₂O and CH₄ from Congo Basin Soils’. All other required amendments fell mostly in line with suggested changes by the reviewer.

SPECIFIC COMMENTS

- Citation 22: Still marked as “in revision” – typo?

Yes, Mendeley typo. Thanks for spotting this.

- Fig. 2: It would be more informative to include smoothed conditional mean lines for lowland vs. montane sites, since you note in line 107 that fluxes were stable/consistent across sites “for each forest type” – ought the reader be able to more clearly distinguish between the forest types, then?

Fig. 2 has been completely remade. The figure now separates the different forest types to better visually distinguish between them.

- Line 120: Might be worth including these data as a supplemental table with simple summary stats, including a measure of variation across the sample.

We have now included the soil temperature and the WFPS data in a new supplementary figure S2 along with some simple summary stats separated by forest type (median, standard deviation). Additionally, summary stats for the fluxes were added via new figures S3 and S4.

- Figure S5 and methods line 299-300: The authors should justify why fluxes with very low R² values were not discarded. Though 80/77% of the N₂O and CH₄ fluxes, respectively,

had good linear fits, Figure S5 indicates that a small proportion of the fluxes have exceedingly poor fits ($R^2 < 0.1$, for instance). E.g., Were these determined to be fluxes close to zero (which are often discarded due to poor flux fits), which weren't discarded in order to not bias the sample towards high fluxes? Is there some other systematic explanation for retaining fluxes in the sample with a poor model fit? The fact that many of the fluxes have a solid fit doesn't on its face justify keeping samples with a poor model fit. Please explain/justify.

We are very grateful that the reviewer is picking up on this and is exactly right with these statements. In our opinion, one should not discard data with poor fits because of the exact problem that one often loses valuable information contained by the very low/zero fluxes which will then bias the data towards high fluxes, as can be nicely seen in our dataset (Fig R1). We were not clear about this in the manuscript and have modified the text to:

‘Note, low R^2 values were mostly associated with low or zero fluxes. Thus, to not bias the data towards high fluxes, none of the data were discarded from analysis.’

Fig R1: Coefficient of determination (R^2) vs. N_2O (left) and CH_4 (right). For illustration purposes, data had been cut off at $0.25 \text{ nmol m}^{-2} \text{ s}^{-1}$ (N_2O) and $\pm 3 \text{ nmol m}^{-2} \text{ s}^{-1}$ (CH_4).

Moreover, we were in the fortunate situation to also have corresponding CO_2 data at hand which gave us further confidence that the seemingly ‘bad’ fluxes were not because of sample handling, contamination, nor instrument issues. That is, only 2.25% of the CO_2 regression coefficient of determination had a value lower than 0.95 and less than 0.5% a regression coefficient lower than 0.6 (Fig R2).

Fig R2: Coefficient of determination (R^2) histograms of each individual regression of CO_2 vs. time.

- It would strengthen the paper to briefly compare (bearing in mind the tight word limit at play, of course) the Fig. 1 data coverage with geographic trends in soil GHG data from other regions (see recommended citations for a big-data approach to this angle).

- Recommended citation: Bond- Lamberty, B., Christianson, D. S., Malhotra, A., Pennington, S. C., Sihi, D., AghaKouchak, A., ... & Zou, J. (2020). COSORE: A community database for continuous soil respiration and other soil- atmosphere greenhouse gas flux data. *Global change biology*, 26(12), 7268-7283.

- Recommended citation: Dorich, C. D., Conant, R. T., Albanito, F., Butterbach-Bahl, K., Grace, P., Scheer, C., ... & van der Weerden, T. J. (2020). Improving N₂O emission estimates with the global N₂O database. *Current Opinion in Environmental Sustainability*, 47, 13-20.

We implemented the suggested references and corresponding databases in the following sentence to stress the poor geographical data coverage compared to other regions.

“While dense greenhouse gas observational infrastructures have been established worldwide at continental scales (e.g. ICOS, NEON) to feed global databases (e.g. FLUXNET (Baldocchi et al., 2001)), ground-based observations from the African continent are still severely underrepresented (D. G. Kim et al., 2016; López-Ballesteros et al., 2018; Valentini et al., 2014) as can be seen from poor sub-Saharan coverage of soil-atmosphere specific greenhouse gas flux databases (COSORE (Bond-Lamberty et al., 2020), Global N₂O Database (Dorich et al., 2020)).”

Reviewer #3 (Remarks to the Author):

This paper is very important as the authors report measurements of nitrous oxide and methane fluxes in a very understudied region of the world: tropical rainforests of the Congo Basin. They find that these forests store more methane and release less nitrous oxide than previously reported or understood. The overall goals of the paper are strong, but the text related to the Methods make it very difficult to assess the validity of the authors' conclusions and the conclusions they draw are too grand based on their sampling design. Here are my main concerns with the paper with detailed comments below:

We thank the reviewer for the very thorough, helpful, and constructive feedback to our manuscript. We have now toned down and away from the extrapolation outcome and deleted these parts from the manuscript. We now report our respective flux magnitudes from the studied forest types and discuss and contextualize them in a wider tropical forest context. Further, we have thoroughly amended and extended the text and information related to the methodology, a point which was also mentioned by the two other reviewers. Lastly, our dataset has expanded as we were able to bring back samples during the revision period of the paper. These new samples were taken during one year at a swamp forest site which we established in 2019. Since ~7% of the Congo Basin's forests are swamp forests we believe that this new data is an important addition to the study. We now cover all major forest types of the basin (montane, lowland, montane) with at least one year of temporal coverage. Despite the addition, we maintained the original manuscript structure. Applied methods and

sampling frequency was identical to the other forest types. Please see below our point-by-point response to the suggested remarks and comments made by the reviewer.

1. If I understand correctly, the authors had a total of five locations per site (as mentioned in line 276) to measure methane and nitrous oxide fluxes, which is too small to make such large conclusions about tropical forests of the Congo Basin as a whole. Instead, I recommend they limit their conclusions to the sites they measured. Soil trace gas fluxes are notoriously heterogeneous, making a sample size of $n = 5$ far too small to justify the grand conclusions made in this paper.

This point was also raised by the two other reviewers. We thus toned down the overall conclusions from our paper and limit them to the sites measured. We furthermore elaborated on the description of the sampling design in the method section and amended Table S1 to reduce the confusion regarding number of sites, measurement frequency, and duration.

2. The study design is very unclear as written, making it difficult to assess the validity of their results. For example, why is there one montane, but three lowland sites included? What distinguishes the long-term from short-term sampling? The authors explain the number of static chambers ($n=5$ per site), but not how apart they are from each other, whether all factors were kept constant across them within a site, or whether they were chosen randomly to capture the heterogeneity of each of the four sites.

We apologize that the method section was confusing. This is partly related to the fact that the sampling was growing and changing over the course of the years. We were trying to condense this information but realize that this led to some general confusion.

We set out for the first time in 2016 to do some initial short term measurements campaigns which took the team to the Yangambi Biosphere Reserve, the Yoko Forest Reserve, the Maringa-Lopori-Wamba-Landscape and to Kahuzi-Biéga National Park. During these short campaigns, a daily sampling frequency was done to assess temporal and spatial flux variability. After these initial campaigns, Yoko Forest Reserve (lowland representative) and Kahuzi-Biéga National Park (montane representative) were chosen to continue long-term sampling, targeting seasonal variability. This is mostly because these were the easiest to reach locations from either Kisangani or Bukavu, which was a pre-requisite for regular field visits. At Yoko Forest Reserve and Kahuzi-Biéga National Park, depending on the year, a weekly or fortnightly sampling frequency was adopted. The swamp forest site, located close to Mbandaka, was added in 2019 to the measurement network as the last missing major forest type.

Lastly, chamber locations were chosen randomly to capture the heterogeneity of the (now) five sites. We included the above summarized information to the paper. The different sections read now:

Introduction

“To achieve this, we quantified seasonal soil-atmosphere fluxes of N_2O and CH_4 from the three major tropical forest types (montane, lowland, swamp) within the Congo Basin at a weekly to fortnightly resolution using the static chamber method (Hutchinson & Mosier, 1981). Initial short-term campaigns with daily measurements were conducted in 2016 and 2017 at three lowland locations (Maringa-Lopori-Wamba Landscape, Yangambi Biosphere

Reserve, Yoko Forest Reserve) and one montane location (Kahuzi-Biéga National Park) to constrain spatiotemporal variation (Fig. S1). Expanding on these short-term expeditions, long-term observation ‘core sites’ were established in Yoko Forest Reserve and Kahuzi-Biéga National Park, as lowland and montane representatives, starting in 2016 and 2017, respectively. These two long-term core sites were subsequently expanded by the addition of a seasonally inundated swamp forest core site in 2019 (Jardin Botanique d’Eala). Including the initial short-term campaigns, 1558 individual soil-atmosphere flux measurements were undertaken for this study (Table S1).”

Methods

“Before the establishment of the long-term observation core sites, two additional lowland forest sites were initially scouted and sampled; one at the Yangambi Biosphere Reserve (Tshopo province) and one in the protected forests surrounding the town of Djolu (Maringa-Lopori-Wamba Landscape, Tshuapa province). Table S1 gives detailed information on location, soil type, sampling duration, replication, periodicity, and total number of fluxes measured at each site.”

3. The authors do not provide the dates of sampling for their soil gas measurements. Knowing whether they overlapped in time with the riverine sampling is essential for determining whether their conclusions comparing soil and riverine data analyses and making conclusions about land to water fluxes.

There was a temporal overlap between terrestrial and riverine sampling. We include now all information related to sampling duration, frequency, and timeframes in Table S1. In addition, we specify this more clearly in the introduction which has now a line implemented:

“...we also report dissolved N_2O and CH_4 concentrations from headwater streams draining the same catchments in which soil flux core sites were located. Stream samples were collected for one year together with the soil flux measurements.”

4. The sample size the authors present in Table S1 is misleading because it provides the total number of times the authors measured fluxes, not the total number of replicates they have per site (which I believe is only 5).

Please see new Table S1.

Specific suggestions:

Lines 42 and 43: Define methane and nitrous oxide before abbreviating

Done.

Lines 45-46: A much better set of numbers to provide here is the total number of replicate chambers per site (which I believe is only 5 per site).

We agree with the reviewer that a more detailed information on the spatial and temporal coverage is needed. We are aware that 5 chambers per site represent a minimum required to be somewhere representative but given the remoteness this was the best we could do. Some locations (Djolu) were only accessible via a 3-day motorbike ride, which meant that all

material had to be transported also by motorbike. This should not be understood as an excuse but rather illustrate that field work conditions were often far from ideal. Moreover, we relocated the chambers occasionally, totaling 6 different chamber locations at the swamp site, 16 different chamber locations at the lowland site and 12 different chambers at the montane site.

We kept the simple numbers of flux measurements in the abstract, as we would like to emphasize to the readership that we have an extensive dataset at hand from which we drew our conclusions but we now rephrased this sentence to be more specific:

“Here, we provide new multi-year data derived from on-ground soil flux ($n = 1558$) and riverine dissolved gas concentration ($n = 332$) measurements spanning montane, swamp, and lowland forests. Each forest type core monitoring site was sampled at least for one hydrological year between 2016 - 2020 at a frequency of 7-14 days.”

Figure 1: This figure is confusing as currently shown. The Abstract states that the authors compiled data from 1267 individual on-ground measurements and 158 riverine dissolved concentrations across the Congo Basin, yet this map shows four sites total in the Congo Basin and it's unclear how the measurements are distributed between on-ground vs. riverine measurements, nor between montane and lowland sites. The authors need to make this map much more informative than what is currently shown.

There are now 3 study sites (1 lowland, 1 montane, 1 swamp) which served as long-term core observation sites for terrestrial fluxes and riverine dissolved gas concentrations. These core sites were complemented by 2 additional lowland terrestrial sites which were initially scouted and 9 additional riverine sites which were sampled only twice (dry and wet season) to assess spatial variability. We amended the map by showing the long-term core observation sites with circles and the supporting sites as squares and made a note that dissolved N_2O and CH_4 concentrations were measured on samples taken from headwater streams draining the same catchments in which core sites were located. The specific information on sampling frequency and duration can now be obtained from Table S1 and the amended Fig1 caption reads now:

“...and this study in yellow (terrestrial) and blue (aquatic). Circles represent core long-term observation sites and squares represent supporting sites. Riverine dissolved CH_4 and N_2O samples were taken from headwater streams draining the same catchments in which the core sites were located.
...”

Moreover, we added a zoomed-in version of Figure 1 to the supplement (Fig S6) to better visualize the location of the supporting riverine sites.

Lines 73-76: The authors bring up the great points of how tropical forests in the Congo Basin are different from forests elsewhere. I suggest they interpret their results in this context, providing much more detail about their sites and how their soils, plants, and other factors may have influenced their results.

We added information about plant, soil and other characteristics to the manuscript, but we refrain from interpreting our results in detail compared to other tropical regions to avoid overinterpretation of our data. Inter-comparisons of the different tropical regions would need a consistent more extensive dataset, covering the pantropics. As such, with this

communication we intend to first and foremost to simply report on-ground fluxes in the Congo Basin and make this dataset available to the wider community.

Line 84: Replace “none date” with “no data”

Done as suggested.

Lines 83-84: Does this one paper correspond to the four sites shown on the map inside the Congo Basin in Figure 1? If not, does it overlap with these four sites?

No, it does not. See new amended Fig 1. This study on CH₄ fluxes (Tathy et al., 1992) was conducted in forested wetlands of the Republic of Congo, located relatively close to our newly added swamp forest site in the DR Congo (close to Mbandaka). We compare the findings of Tathy et al. (1992) to our swamp forest results and now mention that this study was the only one on CH₄ fluxes this far conducted within the confines of the Congo Basin.

Lines 89-90: Here the text sounds like there was just one lowland and one montane forest site. Further down, the authors state that there were measurements taken at “two additional sites” (lines 93-94), but they don’t designate whether these are lowland or montane sites. I had to go down to the Figure 2 legend to understand that there is just one montane forest site (Kahuzi-Biega National Park) and three lowland forest sites. I suggest that the authors make it much clearer earlier in the text (I suggest in lines 89-91) their sites and explain why they had just one montane site, but three lowland sites. Why not an equal distribution of samples? Why not 2 lowland and two montane?

Apart from our earlier comments there are several reasons on why we do not have an equal distribution of forest type sites. The lowland forest type is the most abundant in the Congo Basin (~90%) and therefore it was important to get a better spatial representation. Further, the montane forest type is far less dominant (2.6%) and generally much harder to access to due the steep terrain, especially in the rainy season.

Figure 2: I suggest splitting up the smoothed line to separate the one montane from the three lowland sites. I also suggest adding the actual dates to the X-axis so readers better understand the timing of the study.

We have reworked Figure 2, now separating swamp, montane and lowland forests into different panels. We would like, however, to keep the pooling into weekly classes as the readability of the figure increases and the fluxes are compared on the same relative timelines. We hope that this is fine. We have, however, added a new figure to the Supplement showing all data with corresponding real sampling dates (Fig. S2).

Line 107: The authors state that “For each forest type, N₂O and CH₄ fluxes were stable and consistent within and across sites”, but the way the data are presented in Figure 2 makes it impossible for the reader to decide that for themselves. I suggest, as mentioned above, that the authors present the results of their statistical analyses separately for the four sites. Or at a minimum separate them by the 1 montane vs. 3 lowland forest sites. If the authors are actually referring to the results shown in Figure S1, they should refer to it here.

Figure 2 has now been separated by forest type (but not by sites). Moreover, new figures S3 and S4 show now density distribution of fluxes including summary stats separated by studied forest type, thus specifying distributional information, flux ranges and outlier details.

Lines 122-123: The authors state “Soil N₂O fluxes ranged from -0.108 to 6.42 nmol m⁻² s⁻¹... releasing on average 0.59 kg N₂O-N ha⁻¹ yr⁻¹ from montane and 1.38 kg N₂O-N ha⁻¹ yr⁻¹ from lowland forests.” Again, it’s really hard for the reader to see these differences between sites since the data are not presented separately by forest site.

The data is now presented separately by forest type.

Lines 138-139: Same comments for methane as mentioned above for nitrous oxide.

We have reworked this entire section taking the reviewers comments into account.

Lines 201-214: The authors could describe their forests in much more detail here to try to explain why the results they found are distinct from other tropical forest ecosystems around the globe.

Going into discussing the distinct status of this biome in a pantropical framework would -in our opinion- render a very speculative discussion (and one we feel we should not bring in this piece, given the limited word count). Many factors make this central African rainforest different from other rainforests: the plant community assembly (Slik et al. 2018), the biomass (Lewis et al. 2013), the biogeochemical inputs (Bauters et al. 2018), but also many ‘general’ factors such as soil type distributions (more sandy soils), climate (in general much drier tropical area than South America and South-East Asia) etc. In no way we can account for all these factors, nor do we have a robust field-based pantropical dataset at hand to start exploring with statistical models. Hence, we want to focus on the key message here, which is reporting the fluxes and their variability, and just touching on the contrast with other regions. However, we added here references of published studies done on a subset of our sites which discuss these topics in much more detail (Bauters et al., 2019, Gallarotti et al., 2021). Note, the paper of Gallarotti et al., 2021 was still under revision during our first submission and has been published in the meantime. The sentence reads now:

“However, recent studies from the Congo Basin (Bauters et al., 2019; Gallarotti et al., 2021) and other data on denitrification losses from the Neotropics (Houlton et al., 2006) challenge the paradigm that tropical forests are universal hotspots of N₂O emissions. Instead, these studies rather indicate that a fraction of gaseous N losses occur as N₂ emissions.”

Slik, J. W. F. *et al.* Phylogenetic classification of the world’s tropical forests. *Proc. Natl. Acad. Sci. U. S. A.* **115**, 1837–1842 (2018).

Lewis, S. L. *et al.* Above-ground biomass and structure of 260 African tropical forests Above-gr. *Philos. Trans. R. Soc. Lond. B. Biol. Sci.* **368**, 20120295 (2013).

Bauters, M. *et al.* High fire-derived nitrogen deposition on central African forests. *Proc. Natl. Acad. Sci.* **115**, 549–554 (2018).

Lines 256-271: The methods text is missing a lot of essential information. The paragraph needs a short description of what distinguishes the “long-term observation sites” vs. “short sampling campaigns”. Also, the replication within each of the four sites is very unclear (is it in fact n=5 chambers per site, so n=20 total?). When were the soil gas flux samples taken? The authors explain the dates for stream sampling (March 2018 through February 2019 in line 326), but no such dates are provided for the soil gas measurements (though I see “2016-2019 in the legend for Fig S4). The authors state on line 91 that samples were collected over three years, which three? For how many weeks total? This should be much clearer in the main text of the paper.

We have reworked this section (see above) and hope that together with the implementation of Table S1 and the introduction section, this is clearer now.

Line 305: “To determine site preference” is unclear. The authors need to explain this.

Thank you for pointing this out. The term site preference describes the difference between $\delta^{15}\text{N}$ measured at the central position (alpha) compared to $\delta^{15}\text{N}$ measured at the terminal N atom (beta) of the linear N_2O molecule and is an established term in the N_2O isotope community. Since the article is meant for a broad readership, we have clarified this now writing:

“Moreover, the intramolecular distribution of ^{15}N within the linear N_2O molecule (^{15}N site preference) was analyzed, together with the bulk ^{15}N content ($\delta^{15}\text{N}$) and ^{18}O content ($\delta^{18}\text{O}$) of N_2O , to assess whether N_2O production was dominated by either nitrification or denitrification. To determine ^{15}N site preference, $\delta^{15}\text{N}$ and $\delta^{18}\text{O}$ of surface flux N_2O additional samples were taken during the high frequency sampling campaigns at all sites (Table S1) and analyzed according to a two-source mixing model approach as described in Krüger et al. (2001).”

Line 328-329: The authors state that “To assess spatial variability, several different headwater streams from other catchments close to the flux chamber sites were also sampled..” but they provide no details. They need to include the number of sites total, how far they were from the soil sampling sites, how samples were taken, how statistical analyses were conducted, etc.,

Detailed information about the supporting riverine sites has been added to Table S1 and some more information on the methodology to the text. Moreover, a new zoomed-in version of Fig 1 had been added to the supplement to show the location of these sites (new Fig S6). These additional riverine sites were intended to assess the extent to which the riverine core site data can be generalized over a broader range of streams.

Line 357: The authors state “For the decomposition of variance, we used the complete lowland and montane dataset.” Does this mean they have one montane and three lowland sites in this particular analysis? Were all four sites treated the same way? How is that justified?

We have now changed our statistical analysis approach. Instead of doing a more complex decomposition of variance, we now calculate a coefficient of variation between chambers (within sites) and intra-annual (within sites). This is just to quantify the spatial and temporal variability within sites. For this, we only use the core sites of each forest type (lowland,

montane, swamp). For the quantification of the overall flux, we use an appropriate mixed-effect model per forest type (so three parallel model structures), with random effects for chambers nested in plots, and an additional random effect for sampling date. Including this structure enables the correct weighing of all the different flux measurements, resulting in a true estimate of the fluxes per forest type. The corresponding methods section has now been completely rewritten:

“We used the long-term datasets from Kahuzi-Biéga National Park, Yoko Forest Reserve, and Jardin Botanique d’Eala to assess intra-site (between chamber) and intra-annual variability in the N₂O and CH₄ fluxes. For between chamber variance, we calculated the arithmetic average of the fluxes per chamber over the monitoring periods (see supplementary table S2) and then calculated the variance, and subsequently also the standard deviation and coefficient of variance over those ‘chamber averages’ per site. For the intra-annual variability, an arithmetic average was calculated per week per site (using the different chamber flux estimates). Subsequently, the variance over those weekly site fluxes was calculated, along with the standard deviation and coefficient of variation per site. We estimated median fluxes (from a highly positively skewed distribution of fluxes) for both N₂O and CH₄. This was done by fitting linear mixed effect models per forest type (lowland, montane and swamp forest) with log-transformed fluxes as response variable, no fixed effects, and chamber ID nested within plot as a random intercept, with an additional random intercept for sampling date. The effect estimates for the intercept of the different models, corroborating with a ‘best estimator’ for the fluxes in the different forest types, were exponentiated for the estimation of median fluxes per forest type”

Lines 374-375: The authors note that “soil and riverine data collected were pooled into weekly classes irrespective of year only for visualization purposes”, but they didn’t provide the years for when soils were sampled. Were the soils sampled at the same time as river samples?

We refer to the new Table S1 and Fig S2 and comments given above.

Figure S1, S2, S4: I suggest adding the actual years to the X-axis of all three figures so readers better understand the timing of the study.

Figure S1: Added years as suggested.

Figure S2: This figure has now moved to the main part of the manuscript. We would like to keep the pooling into weekly classes as the readability of the figure increases and the fluxes and dissolved gas concentrations can be compared on the same relative timelines.

Former Fig S4 is now part of new Fig S2 showing daily sums of individual chamber measurements separated by site together with the actual flux measurements as well as soil temperature and soil water filled pore space over the entire duration of the study.

Figure S3: A map with these sites would strengthen the paper. At a minimum, the distance between these sites, the dominant tree species, soil types, etc., are needed to better understand how well these sites represent the region.

A zoomed-in version of Fig 1 with locations of supporting riverine sampling sites (new Fig S6) is now added to the supplementary information together with more details on soil type in

Table S1. Since these additional sites were relatively close to our core sites we assume that tree species assemblage is similar in these catchments.

Figure S3: What is the sample size for each of these sites? It is difficult to assess these results without that. I don't mean the number of times they measured fluxes, I mean the number of spatial replicates utilized.

See earlier comments.

Reviewers' Comments:

Reviewer #1:

Remarks to the Author:

I find the article improved, my suggestions were addressed and I don't have further comment.

Reviewer #2:

Remarks to the Author:

The authors' responses to reviewer comments read to me as thorough and detailed and, I believe, substantially improve this manuscript.

My primary critiques after initial submission concerned (a) my skepticism of whether the spatial scaling exercise was sufficiently rigorous and appropriate based on the collected data, and (b) several points of confusion around field and analytical methods, both of which were brought up by the other reviewers.

For the first, the authors removed the extrapolation exercise (both results and text) from the manuscript. I support this decision and think that the paper is strengthened by being refocused on its core data and findings. The authors did add an alternate extrapolation analysis, in which they calculate the weighted average soil flux of CH₄ and N₂O (weighted by spatial coverage of the three forest types in the Congo Basin), found in lines 182 (CH₄) and 198 (N₂O). This analysis would be improved if a measure of uncertainty were included (SD, SE or 95% confidence interval).

For the second, the authors rewrote portions of the text throughout, including in the abstract, and added text to the methods sections. Overall, I think the methods are much more understandable in this iteration of the manuscript. I also think that Table S1 and Table S2 are welcome changes that improve transparency and readability.

Broadly, I think this paper has improved substantially and that the numerous confusing/unclear explanations of the methods (replicate numbers, site locations, scaling exercise, etc.) have been resolved. I also think that removing the extrapolation exercise and focusing the manuscript on the core findings that come from the authors' data collection is appropriate.

I am happy with the extent to which my other, more minor comments, were addressed.

Reviewer #4:

Remarks to the Author:

Comments to the manuscript by Barthel et al, NCOMMS-20-50731A, "Low N₂O and variable CH₄ fluxes from tropical forest soils of the Congo Basin".

General comment:

I have been asked to review the current version of the ms. As I was not part of the initial review round, I have also been asked to assess whether the authors have addressed the comments by the initial reviewer #3 in a satisfactory way.

The manuscript focus on an important topic that I believe is suitable for publication in Nature Communications. Echoing the other reviewers, the Congo Basin is a heavily understudied system from multiple perspectives, not at least when it comes to the cycling of radiative important trace gases such as CH₄ and N₂O. Hence, the current study is a timely and very important contribution to the research

field. The manuscript is highly interesting and well-written, and from what I can judge the authors have made a good job in responding to the comments by reviewer #3 of the original review round. One of the main comments was about "over-generalizing" the results of the study which I think in the revised version is toned down and now more balanced. The authors have in most cases followed the suggestions by the reviewer, but also in a convincing and clear way motivated when they have not followed the critic. Despite that I think the authors have done a great job with the revisions according to the earlier comments I have some additional concerns on how suitable it is for publication in its existing form.

My main concern is how the terrestrial-aquatic connection in the ms is described and interpreted. I appreciate very much that the authors, in addition to soil exchange, also have sampled for CH₄ and N₂O in connecting headwater streams. However, the authors claim in the header on Ln200 that they "linking terrestrial and aquatic fluxes". I find this partly misleading as there are no aquatic fluxes measured or calculated in the ms, instead the authors relate terrestrial fluxes with aquatic concentrations, which to some extent is comparing apples with pears. Also, there are numerous of factors (in addition to soil export) that affect the in-stream concentrations of CH₄ and N₂O (hydrological conditions, stream channel morphology, stream channel gradient etc.) which affect both the terrestrial and aquatic source areas for instream CH₄ and N₂O, the ability for "consumption" of the two gases, as well as the conditions for air-water exchange. Also, both CH₄ and N₂O are poorly soluble in water and both gases are well-known to be highly variable at low spatiotemporal scales in stream networks. So although I agree with much of the reasoning that the authors make in the section Ln 200-230 I think they need to be much more transparent with what they compare and that measuring stream concentrations of a certain gas do not necessarily reflect lateral soil export from the spot where the soil-atmosphere fluxes were measured. For example, that N₂O in montane and lowland stream have near equilibrium concentrations of N₂O and close to atmospheric delta15N do not (in my mind) necessarily need to mean that "negligible quantities of N₂O are dissolved in soil pore water and laterally exported to aquatic systems". The N₂O might already be emitted to the atmosphere before being sampled due turbulent upstream conditions. Also, despite low concentrations high precip/runoff could still result in that significant quantities of N₂O are exported from soil to water. I recommend that the authors make this section more nuanced and clear in that there is no comparison between vertical and lateral fluxes, and that the approach of using stream concentrations to interfere processes and connectivity with soil-atmosphere fluxes is likely associated with high uncertainty.

Minor comments

Figure 1. The color legend in the map is hard to read in the version I have. Just a thin light green line at 100% and the rest is white. Although it is rather self-describing in the map I suggest to improve the legend.

Ln 417-427, From the method text I cannot see that any kind of preservation of the dissolved N₂O samples taken for isotopic analysis was made. Was this not done and if so, I assume this would have caused fractionation within the samples due to various processes which in turn affect the isotopic composition being analyzed? I.e. an unknown discrepancy between field and lab delta15N?? Please clarify

Reviewer #1 (Remarks to the Author):

I find the article improved, my suggestions were addressed and I don't have further comment.

We thank the reviewer for their positive feedback on our revised version.

Reviewer #2 (Remarks to the Author):

The authors' responses to reviewer comments read to me as thorough and detailed and, I believe, substantially improve this manuscript.

My primary critiques after initial submission concerned (a) my skepticism of whether the spatial scaling exercise was sufficiently rigorous and appropriate based on the collected data, and (b) several points of confusion around field and analytical methods, both of which were brought up by the other reviewers.

For the first, the authors removed the extrapolation exercise (both results and text) from the manuscript. I support this decision and think that the paper is strengthened by being refocused on its core data and findings. The authors did add an alternate extrapolation analysis, in which they calculate the weighted average soil flux of CH₄ and N₂O (weighted by spatial coverage of the three forest types in the Congo Basin), found in lines 182 (CH₄) and 198 (N₂O). This analysis would be improved if a measure of uncertainty were included (SD, SE or 95% confidence interval).

We thank the reviewer for their suggestion to add a measure of uncertainty to our extrapolation analysis. This is an excellent suggestion and is now implemented into the paper by additionally weighting the upper and lower confidence interval boundaries the same way we did for the weighted average.

For the second, the authors rewrote portions of the text throughout, including in the abstract, and added text to the methods sections. Overall, I think the methods are much more understandable in this iteration of the manuscript. I also think that Table S1 and Table S2 are welcome changes that improve transparency and readability.

We are glad that these changes and implementations were well received. Thank you.

Broadly, I think this paper has improved substantially and that the numerous confusing/unclear explanations of the methods (replicate numbers, site locations, scaling exercise, etc.) have been resolved. I also think that removing the extrapolation exercise and focusing the manuscript on the core findings that come from the authors' data collection is appropriate.

I am happy with the extent to which my other, more minor comments, were addressed.

We thank the reviewer once more for their excellent and thorough feedback on our manuscript which has certainly improved the overall quality of our work.

Reviewer #4 (Remarks to the Author):

Comments to the manuscript by Barthel et al, NCOMMS-20-50731A, “Low N₂O and variable CH₄ fluxes from tropical forest soils of the Congo Basin”.

General comment:

I have been asked to review the current version of the ms. As I was not part of the initial review round, I have also been asked to assess whether the authors have addressed the comments by the initial reviewer #3 in a satisfactory way.

The manuscript focus on an important topic that I believe is suitable for publication in Nature Communications. Echoing the other reviewers, the Congo Basin is a heavily understudied system from multiple perspectives, not at least when it comes to the cycling of radiative important trace gases such as CH₄ and N₂O. Hence, the current study is a timely and very important contribution to the research field. The manuscript is highly interesting and well-written, and from what I can judge the authors have made a good job in responding to the comments by reviewer #3 of the original review round. One of the main comments was about “over-generalizing” the results of the study which I think in the revised version is toned down and now more balanced. The authors have in most cases followed the suggestions by the reviewer, but also in a convincing and clear way motivated when they have not followed the critic. Despite that I think the authors have done a great job with the revisions according to the earlier comments I have some additional concerns on how suitable it is for publication in its existing form.

We greatly appreciate the positive feedback on our manuscript and the positive assessment that we addressed the initial reviewer’s concerns in a satisfactory way.

My main concern is how the terrestrial-aquatic connection in the ms is described and interpreted. I appreciate very much that the authors, in addition to soil exchange, also have sampled for CH₄ and N₂O in connecting headwater streams. However, the authors claim in the header on Ln200 that they “linking terrestrial and aquatic fluxes”. I find this partly misleading as there are no aquatic fluxes measured or calculated in the ms, instead the authors relate terrestrial fluxes with aquatic concentrations, which to some extent is comparing apples with pears. Also, there are numerous of factors (in addition to soil export) that affect the in-stream concentrations of CH₄ and N₂O (hydrological conditions, stream channel morphology, stream channel gradient etc.) which affect both the terrestrial and aquatic source areas for instream CH₄ and N₂O, the ability for “consumption” of the two gases, as well as the conditions for air-water exchange. Also, both CH₄ and N₂O are poorly soluble in water and both gases are well-known to be highly variable at low spatiotemporal scales in stream networks. So although I agree with much of the reasoning that the authors make in the section Ln 200-230 I think they need to be much more transparent with what they compare and that measuring stream concentrations of a certain gas do not necessarily reflect lateral soil export from the spot where the soil-atmosphere fluxes were measured. For example, that N₂O in montane and lowland stream have near equilibrium concentrations of N₂O and close to atmospheric delta¹⁵N do not (in my mind) necessarily need to mean that “negligible quantities of N₂O are dissolved in soil pore water and laterally exported to aquatic systems”. The N₂O might already be emitted to the atmosphere before being sampled due turbulent upstream conditions. Also, despite low concentrations high precip/runoff could still result in that significant quantities of N₂O are exported from soil to water. I recommend that the authors make this section more nuanced and clear in that there is no

comparison between vertical and lateral fluxes, and that the approach of using stream concentrations to interfere processes and connectivity with soil-atmosphere fluxes is likely associated with high uncertainty.

We greatly appreciate the reviewer for pointing this out. These are important points and we agree that this comparison is currently inappropriate and the title of the section was therefore misleading. We have now added more nuance to the discussion by incorporating/reiterating the reviewer's excellent comments and renaming the section to: 'Linking terrestrial (vertical) and aquatic (lateral) N₂O and CH₄'. The added paragraph now reads as follows:

It is important to note that in-stream gas concentrations are a result of a multitude of simultaneous processes and drivers (i.e. gas solubility, aquatic-terrestrial connectivity, inputs from surface water/groundwater, in-stream metabolism, stream chemistry, stream morphology, gas transfer velocity) which render CH₄ and N₂O highly variable at low spatiotemporal scales. Thus, although the in-stream concentrations we observed are consistent with minimal vertical N₂O losses, decoupled CH₄ fluxes in the lowland and montane, and strong hydrologic connectivity in the swamp forest, they do not necessarily reflect lateral soil export from the location in the catchment where soil-atmosphere fluxes were measured. We therefore caution that such terrestrial-aquatic linkage is likely associated with high uncertainty.

Minor comments

Figure 1. The color legend in the map is hard to read in the version I have. Just a thin light green line at 100% and the rest is white. Although it is rather self-describing in the map I suggest to improve the legend.

Since the forest cover legend shows a gradient from white to green which is difficult to improve we decided to entirely remove it from the figure as it is indeed self-describing and added this information to the figure caption instead. It reads now:

'2019 forest cover of sub-Saharan Africa (0% white, 100% dark green) with Congo Basin boundary delineation...'

Ln 417-427, From the method text I cannot see that any kind of preservation of the dissolved N₂O samples taken for isotopic analysis was made. Was this not done and if so, I assume this would have caused fractionation within the samples due to various processes which in turn affect the isotopic composition being analyzed? I.e. an unknown discrepancy between field and lab delta¹⁵N?? Please clarify

Thank you for pointing out this important detail which we forgot to mention. The vials used for isotopic analysis were also pre-treated with ZnCl₂ in order to suppress microbial activity after sample injection. We have added now this information to the materials and method section:

‘Similar to the preparation of the Labco exetainers, 110 mL serum crimp vials used for the isotopic characterization of dissolved N₂O were pre-flushed with ultrapure N₂ and pre-treated with 200 μL of 50 % (w/v) ZnCl₂ to stop microbial activity after sample injection’